# New insights on the South China Sea Throughflow and water budget seasonal cycle: evaluation and analysis of a high-resolution configuration of the ocean model SYMPHONIE version 2.4

Ngoc B. Trinh[1,2], Marine Herrmann[1,2], Caroline Ulses[1,2], Patrick Marsaleix[1], Thomas Duhaut[1], To Duy Thai[3], Claude Estournel[1], R. Kipp Shearman[4]

[1]Université de Toulouse, LEGOS (IRD/CNES/CNRS/UT3), 31400 Toulouse, France

[2]LOTUS Laboratory, University of Science and Technology of Hanoi (USTH), Vietnam Academy of Science and Technology (VAST), Hanoi, Vietnam

[3]Institute of Oceanography (IO), Vietnam Academy of Science and Technology (VAST), Nha Trang, Vietnam

[4]College of Earth, Ocean, and Atmospheric Sciences, Oregon State University, Corvallis, OR United States

*Correspondence to*: Ngoc Bich Trinh (trinh-bich.ngoc@usth.edu.vn)

**Short summary**

A high-resolution model was built to study the South China Sea (SCS) water, heat and salt budgets. The model
performance is demonstrated by comparison with observations and other simulations. Important discards are observed if calculating offline, instead of online, lateral inflows and outflows of water, heat and salt. SCS mainly receives water from the Luzon Strait and releases it through Mindoro, Taiwan and Karimata Straits. SCS surface interocean water exchanges are driven by monsoon winds.

**Abstract**

The South China Sea Throughflow (SCSTF) connects the South China Sea (SCS) with neighboring seas and oceans, transferring surface water of the global thermohaline circulation between the Pacific and Indian oceans. A configuration of the SYMPHONIE ocean model at high resolution (4 km) and including an explicit representation of tides is implemented over this region, and a simulation is analyzed over 2010-2018. Comparisons with in-situ and satellite data and other available simulations at coarser resolution show the good performance of the model and the
relevance of the high resolution in reproducing the spatial and temporal variability of the characteristics of surface dynamics and water masses over the SCS. The added value of an online computation of each term of the water, heat and salt SCS budgets (surface, lateral oceanic and river fluxes and internal variations) is also quantitatively

demonstrated: important discards are obtained with offline computation, with relative biases of ~40 % for lateral oceanic inflows and outflows.

The SCS water volume budget, including the SCSTF, is analyzed at the climatological and seasonal scales. The SCS receives on average a 4.5 Sv yearly water volume input, mainly from the Luzon Strait. It laterally releases this water to neighbouring seas, mainly to the Sulu Sea through Mindoro Strait (49 %), to the East China Sea via Taiwan Strait (28 %) and to the Java Sea through Karimata Strait (22 %). The seasonal variability of this water volume budget is driven by lateral interocean exchanges. Surface interocean exchanges, especially at Luzon Strait, are all driven by

monsoon winds which favor winter southwestward flows and summer northeastward surface flows. Exchanges through Luzon Strait deep layers show a stable sandwiched structure with vertically alternating inflows and outflows. Last, differences in flux estimates induced by the use of a high vs. low resolution model are quantified.

## 1. Introduction

The South China Sea (SCS, Fig. 1a), the largest marginal sea in the world, is subject to a wide range of forcings at

different scales of both natural and anthropic origins. Its coasts are among the most densely populated regions in the world (CIESIN, 2018). The SCS is a source of subsistence for these populations (fishing, tourism, etc.) and is reciprocally affected by the harmful effects of human activities (pollution, resources overexploitation, etc.). The SCS plays an important role in regional and global ocean circulation and climate, transferring the surface water masses of the global thermohaline circulation between the Pacific and Indian oceans (Qu et al. 2005; Tozuka et al. 2007). It is

therefore essential to understand, quantify and monitor the respective contributions of the lateral oceanic, atmospheric and continental fluxes in the SCS water, heat and salt budgets, and their interactions.

Ocean dynamics drive the transport and mixing of water masses and are thus strongly involved in the functioning and variability of the water, heat and salt budgets of the SCS. They also determine the fate and functioning of matter in the marine compartment (planktonic ecosystems, contaminants, sediments). The SCS ocean circulation is regulated

by a combination of factors, including the geometry of the zone, the tides, the connection with the Western Pacific and Eastern Indian oceans and the atmospheric forcing, from the daily to the seasonal and interannual scales (Wyrtki, 1961; Shaw and Chao, 1994; Metzger and Hurlburt, 1996; Gan et al. 2006). In the upper layer, the SCS basin scale circulation is mainly driven by the seasonal monsoon winds (Liu et al. 2002; Liu and Gan, 2017). In winter, strong northeasterly monsoon winds generate a cyclonic circulation in the surface and upper layers over the whole basin. In

summer, weaker southwesterly monsoon winds lead to a cyclonic gyre in the north and an anticyclonic gyre in the south (Qu, 2000; Gan et al. 2016). At the interannual timescale, the SCS circulation is impacted by the El Niño Southern Oscillation (ENSO), via its effect on monsoon winds (Soden et al. 1999; Liu et al. 2014; Tan et al. 2016) but also via the direct propagation of ENSO oceanic signals from the Western Pacific Ocean through the Luzon Strait (Qu et al. 2004; Wang et al. 2006a). Other studies also suggested an impact of the Pacific Decadal Oscillation (PDO) on

the SCS related to its effect on the intrusion of Western Pacific water (Yu and Qu, 2013) and on the atmospheric water

flux (Zeng et al. 2018). On the other side of the spectrum, the SCS is frequently crossed by tropical cyclones (Wang et al. 2007) that also affect ocean dynamics (Pan and Sun, 2013) and ecosystems (Liu et al. 2019). Last but not least, mesoscale to submesoscale structures play a significant role in the water masses dynamics and transports within the SCS (Liu et al. 2008; Chen et al. 2011; Nan et al. 2015; Da et al. 2019; Lin et al. 2020; Ni et al. 2021; Herrmann et al. 2023).

The SCS is connected with surrounding oceans and seas by several straits (Fig. 1a, white lines). The sills of Luzon and Mindoro Straits are 3000 m and 400 m deep respectively, the other straits are less than 100 m deep. The Luzon Strait – the largest and deepest interocean strait of the zone – is the main pathway of seawater from the Pacific Ocean into the SCS (Wyrtki, 1961). Besides, the SCS exchanges seawater with the East China Sea through the Taiwan Strait, with the Sulu Sea through the Straits of Balabac and Mindoro, with the Java Sea and Andaman Sea (Indian Ocean) through Karimata and Malacca Straits. Mindoro, Balabac and Malacca Straits are particularly narrow, with widest passages not wider than 80, 55 and 20 km respectively. Based on numerical studies, satellite observations and long-term wind data analyzes, Qu et al. (2005) and Yu et al. (2007) revealed a circulation where Pacific Ocean water masses enter the SCS through the Luzon Strait and leave the basin through Taiwan, Karimata and Mindoro Straits, forming the South China Sea Throughflow (SCSTF). Those lateral transports are involved in the SCS cycle of water, heat and salt and interact with the atmospheric and continental components of this regional cycle. The SCS indeed receives net gains of freshwater and heat from the atmosphere and rivers. Estimates of net surface heat gain vary from 17 to 51 W m$^{-2}$ (Yang et al. 1999; Qu et al. 2004; Yu and Weller, 2007; Fang et al. 2009; Wang et al. 2019) and estimates of net water gain vary between 0.05 and 0.2 Sv (Qu et al. 2006; Fang et al. 2009).

Previous estimates of water volume, heat and salt transports at the straits were performed based on in-situ and satellite observations (Fang et al. 1991; Chu and Li, 2000; Chung et al. 2001; Wang et al. 2003; Tian et al. 2006; Yuan et al. 2008; Fang et al. 2010; Qu and Song, 2009; Sprintall et al. 2012; Susanto et al. 2013). However, in-situ estimates remain limited in space and time and are made complicated by the complex topography in the region. Numerical modeling is a relevant tool to complement in-situ and satellite measurements. Several modeling studies based on an integrated approach considering all terms of the budgets were performed, mainly focusing on water volume fluxes. Yaremchuk et al. (2009) provided estimates of upper water volume transport at Luzon, Taiwan, Mindoro and Karimata Straits issued from a reduced - gravity model. Wang et al. (2009), using a ~18 km resolution model, evaluated the seawater fluxes through all SCS interocean straits. In both studies, the inflow at Luzon was considered to be balanced by the outflows at other straits, i.e., internal variations were neglected, and the contribution from the atmosphere and rivers was not considered. Liu et al. (2011), Hsin et al. (2012), Tozuka et al. (2015), Wei et al. (2016) provided estimates of the SCS interocean water volume transports with higher resolution numerical models, but models configurations and assumptions did not allow to rigorously close the water volume budget. Several studies addressed the question of heat and salt fluxes. Qu et al. (2004) studied the whole depth water volume transports through Luzon, Mindoro and Sunda Straits and the upper heat budget of the zone, revealing that the surface heat flux is the primary heating process. However, their numerical study was carried out with a closed Taiwan Strait and a shallower Mindoro

Strait than reality, the inflow at Luzon was balanced by outflows at Mindoro and Sunda Straits, and the river heat flux was neglected. Qu et al. (2006), using a ~11 km resolution model, estimated the total water volume, heat and freshwater SCSTF, deducing surface heat and freshwater transports from the difference between the inflowing and outflowing fluxes of temperature and salinity. Fang et al. (2005, 2009) were the first, followed by Wang et al. (2019),

to evaluate transports through all interocean straits of the SCS, using respectively ~18 km then ~7 km resolution models, but assuming that outflows compensate for inflows.

Those studies considerably improved our understanding of water volume, heat and salt transport through the SCS area. However, they were associated with several limitations. First, they assumed that the SCS is at equilibrium over the studied periods, i.e., that the same amount of water volume, heat and salt that enters the basin leaves it, and used this

assumption to deduce atmosphere and rivers contributions. Though this assumption allows to close the budget at the first order, it does not account for possible internal variations and trends in the water volume, heat and salt contents of the SCS. Yet Zeng et al. (2014, 2018), using in situ measurements and satellite data, evidenced a freshening of the SCS from 2010 to 2012 followed by a salinification until 2016, suggesting an interannual variability in salt and/or water mass content. Moreover, very few studies examined jointly the water volume, heat and salt budgets, which is

however necessary to provide consistent estimates of all the terms involved in those budgets and understand their interactions. Besides, using available (re)analysis to study those budgets requires to compute them offline, based on daily, weekly or even monthly distributed outputs, thus neglecting the turbulent term of temperature and salinity lateral transports. The error associated with this assumption requires to be assessed. Last, the model's resolution was rarely finer than 10 km, and they did not represent tides. As pointed out by Lin et al. (2020), models at higher resolution and

including tides are necessary to represent the full range of temporal and spatial scales involved in the transport and mixing of water masses through the SCS. This includes the mesoscale to submesoscale eddies and structures of size smaller than 40 km (Da et al. 2019, Ni et al. 2021, Herrmann et al. 2023), as well as the detailed topography and dynamics of coastal areas and key straits, some of which less than 20 km wide, where interocean exchanges and strong internal tidal mixing occur (e.g. at Luzon Strait but also at narrow straits like Malacca, Mindoro and Balabac,

Hatayama et al. 1996, Laurent 2008, Wang et al. 2016). Xu et al. (2022) indeed showed for the Atlantic Ocean that using in a model combining a high resolution (1/50°) bathymetry and explicit tides improved the representation of internal waves, and consequently of the mesoscale sea surface height wavenumber spectrum over the tropical ocean. Sannino et al. (2009) and Chassignet et al. (2023) moreover pointed out the relevance of high-resolution bathymetry for the representation of interocean strait exchanges and mesoscale activity involved in western boundary currents,

respectively.

Following this introduction, our first scientific objective is to better understand the role of the SCS in the global circulation and regional climate at different scales, i.e., daily, seasonal and interannual variability, by providing updated and consistent estimates at those scales of all the terms involved in the SCS water volume, heat and salt budgets: lateral oceanic, atmospheric and river fluxes and internal variations. For that, we developed a configuration

of a regional ocean hydrodynamical model with a high spatial resolution (4 km) over the SCS and an explicit

representation of tides, in order to represent as realistically as possible the wide range of scales and processes involved in the SCS dynamics and to study their contribution to SCS budgets. The water volume, heat and salt budgets have been rigorously closed by performing online calculations of each term of those budgets, including incoming and outgoing flows. The first objective of this paper is to present and evaluate in detail this modeling tool, that will be used to study water volume, salt and heat budgets, and will be available to the community interested in addressing scientific questions related to SCS ocean dynamics functioning, variability and influence. The second objective is to perform a first analysis at the climatological and seasonal scales of water budget over the SCS and of its components, i.e. river, atmospheric and oceanic lateral fluxes and internal variations.

The paper is organized as follows. Section 2 presents the hydrodynamical model, its high-resolution configuration over the SCS and the observation data as well as the other numerical simulations at coarser resolution used for its evaluation. The online computation of each term of the budgets are then detailed. The added-value of the online computation compared to the offline computation is demonstrated in Section 3. The ability of the model to simulate the SCS sea surface dynamics and water masses characteristics at different scales is evaluated in Section 4 through comparisons with available in-situ and satellite observations and with other simulations. An evaluation and an analysis of the water budget and its various components over the SCS are carried out on a climatological scale in Section 5, and lateral exchanges at interocean straits, corresponding to the SCSTF, are examined in detail. Results are summarized in Section 6, and an overview on the future applications of this high-resolution closed-budget modeling tool is provided.

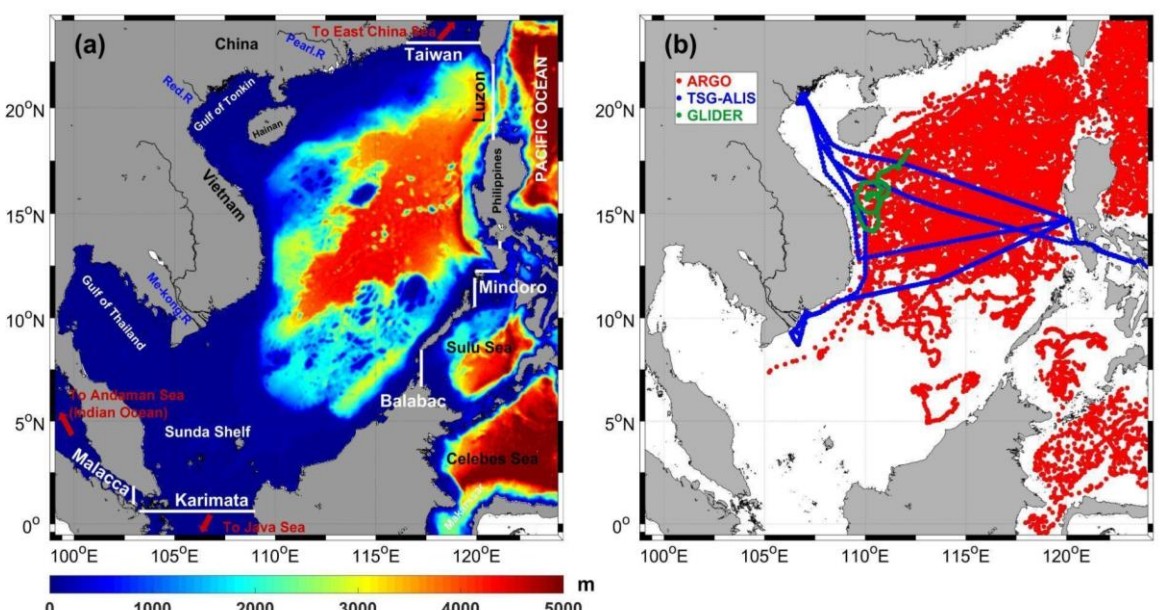

**Figure 1: (a) Computational domain bathymetry (m) and interocean straits (white lines). (b) Maps of Argo float trajectories in the SCS from January 2009 to December 2018 (red), TSG from R/V Alis trajectory from May to July 2014 (blue), and Glider trajectory from January to May 2017 (green).**

## 2. Materials and methods

### 2.1 The numerical model SYMPHONIE


#### 2.1.1 General presentation of the model

The 3-D ocean circulation model SYMPHONIE Marsaleix et al. (2008, 2019) is based on the Navier-Stokes primitive equations solved on an Arakawa curvilinear C-grid under the hydrostatic and Boussinesq approximations. The model makes use of an energy conserving finite difference method (Marsaleix et al. 2008), a forward-backward time stepping scheme, a Jacobian pressure gradient scheme (Marsaleix et al. 2009), the equation of state of Jackett et al. (2006), and

the K-epsilon turbulence scheme with the implementation described in Costa et al. (2017). Horizontal advection and diffusion of tracers are computed using the QUICKEST scheme (Leonard, 1979) and vertical advection using a centered scheme. Horizontal advection and diffusion of momentum are each computed with a fourth order centered biharmonic scheme. The biharmonic viscosity of momentum is calculated according to a Smagorinsky-like formulation derived from Griffies and Hallberg (2000). The lateral open boundary conditions, based on radiation

conditions combined with nudging conditions, are described in Marsaleix et al. (2006) and boundary conditions at river mouths are described in Nguyen-Duy et al. (2021). As in Estournel et al. (2021), To Duy et al. (2022) and Hermann et al. (2023), the VQS (vanishing quasi-sigma) vertical coordinate is used, allowing to avoid an excess of vertical levels in very shallow areas while maintaining an accurate description of the bathymetry and to reduce the truncation errors associated with the sigma coordinate.

#### 2.1.2 Model setup

The SYMPHONIE numerical configuration covers the whole SCS (99°E – 124°E, 0.6°S – 24°N, Fig. 1a), with a regular grid of 4 km horizontal resolution and 50 vertical levels in the deepest area. It is built from a bathymetry product merging GEBCO 2014 gridded bathymetry and digitalized nautical charts (Piton et al. 2020). Bathymetry ranges from 3 m to 5000 m in the studied area (Fig. 1a). The simulation runs from 01 January 2009 to 31 December

2018 and is referred to as SYM4 in the following.

Initial and lateral oceanic boundary conditions for temperature, salinity, currents and sea level are provided by the daily outputs of the Global_Analysis_Forecast_Phy_001_024 Global Ocean 1/12° physics analysis and forecast provided by Copernicus Marine Environment Monitoring Service (CMEMS) (http://marine.copernicus.eu; last access 18 May 2023). The SYM4 simulation departs from an initial state that is not at rest since it includes the currents from

CMEMS 1/12° analysis. The spin-up time, whose main aim is to energetically adjust the initial physical fields provided by CMEMS to the specific constraints of the SYM4 grid, lasts a few months. We therefore analyze the simulation over the period January 01, 2010 - December 31, 2018.

The SCS configuration includes 63 river mouths. Daily data were provided by the National Hydro-Meteorological Service of Vietnam for 11 rivers flowing in northern and central Vietnam (including the Red river). Monthly

climatology runoff issued from the CLS – INDESO project were provided for the other rivers, including the Mekong and Pearl rivers (Tranchant et al. 2016).

The atmospheric forcing is calculated from the bulk formulae of Large and Yeager (2004) using the European Centre for Medium Range Weather Forecasts (ECMWF) operational forecasts at 1/8° horizontal resolution and 3 hours temporal resolution, available at https://www.ecmwf.int/, last access 18 May 2023.

Open boundary tidal conditions are prescribed from FES2014b, the 2015 release of the FES (Finite Element Solution) global tide model (Carrere et al. 2016, Lyard et al., 2021), that assimilates altimetry satellite observations and tide gauges data. The data are freely available on Aviso website: https://www.aviso.altimetry.fr/en/data/products/auxiliary-products/global-tide-fes.html (last access 18 May 2023). The SCS configuration takes into account nine barotropic tidal components (in phase and altitude): M2, S2, N2, K2

(semi-diurnal tides), K1, P1, O1, Q1 (diurnal tides) and M4 (compound tide). The model is also forced by the astronomical plus the loading and self-attraction potentials (Lyard et al. 2006). Details and numerical issues related to tides can be found in Pairaud et al. (2008, 2010).

**2.2 Fluxes calculation methods**

We detail here the computation of each term of the water volume, heat and salt budgets over the whole SCS: internal
content variations and surface, oceanic lateral and river fluxes. We compute lateral oceanic fluxes through the six interocean straits connecting the SCS to neighboring seas and oceans shown in Fig. 1a: Taiwan, Luzon, Mindoro, Balabac, Karimata and Malacca Straits. All the terms of the budget equations are computed online. The added-value of the online computation compared to the offline computation is presented in Section 3.

**2.2.1    Water volume, heat and salt budgets equations**

Water volume, heat and salt contents are rigorously conserved in SYMPHONIE, as shown below in Section 3 and Fig. 2: during each time step the variation (delta) of water volume, heat or salt content in the numerical ocean domain is equal to the net input from sources and sinks, i.e. the sum of fluxes from rivers, atmosphere and lateral oceanic boundaries.

***Water volume budget***

The internal variation of water volume V over the SCS area between times $t_1$ and $t_2$ ($\Delta V$) is equal to the integral between $t_1$ and $t_2$ of all water volume fluxes exchanged at the boundaries (atmosphere, rivers and lateral open ocean boundaries) of the SCS domain, taken as the sea zone limited by the six interocean straits shown in Fig. 1a:

$$\Delta V = V_{t2} - V_{t1} = \int_{t_1}^{t_2} \big( F_{w,lat} + F_{w,surf} + F_{w,riv} \big) dt \qquad (Eq.1)$$

where $F_{w,lat}$, $F_{w,surf}$ and $F_{w,riv}$ are the net oceanic lateral, atmospheric surface and river water volume fluxes respectively. Here and in the following, positive fluxes correspond to inflows, and negative fluxes to outflows.

*Heat budget*

The variation of heat content $HC$ between times $t_1$ and $t_2$ ($\Delta HC$) is equal to the sum of all heat fluxes exchanged at the boundaries of the SCS domain between $t_1$ and $t_2$:

$$\Delta HC = HC_{t2} - HC_{t1} = \int_{t_1}^{t_2} \left( F_{T,lat} + F_{T,surf} + F_{T,riv} \right) dt \qquad (Eq.\,2)$$

where $F_{T,lat}$, $F_{T,surf}$ and $F_{T,riv}$ are the net oceanic lateral, atmospheric surface and river heat fluxes respectively, and HC is computed from:

$$HC = \rho_0 C_p \int_x \int_y \int_z T(x,y,z,t) dx\, dy\, dz \qquad (Eq.\,3)$$

with $T$ the temperature (in °C), $\rho_0$ the seawater density constant (1028 kg m$^{-3}$), $C_p$ the seawater specific heat constant (3900 Jkg$^{-1}$°C$^{-1}$).

*Salt budget*

The salinity of water going to or coming from to the atmosphere and the rivers is assumed to be zero, meaning that there is no input or output of salt from surface atmospheric fluxes and river runoff. It should be noted that evaporation, precipitation and river discharge are not sources/sinks of salt but are however sources/sinks of salinity for the ocean domain: although they do not affect the salt budget of the ocean domain, atmospheric and river fluxes do modify the salinity budget, as they affect the water volume budget. The variation of salt content between $t_1$ and $t_2$ ($\Delta SC$) is thus equal to the sum of salt fluxes exchanged at the lateral oceanic boundaries of the SCS domain:

$$\Delta SC = SC_{t2} - SC_{t1} = \int_{t_1}^{t_2} F_{S,lat} dt \qquad (Eq.\,4)$$

where $F_{S,lat}$ is the net salt flux at the lateral oceanic boundaries and SC is computed from:

$$SC = \rho_0 \int_x \int_y \int_z S(x,y,z,t)\, dx\, dy\, dz \qquad (Eq.\,5)$$

with S the salinity.

**2.2.2 Lateral fluxes through ocean open boundaries**

The total lateral water volume flux $F_{w,lat}$ through the vertical section A of an open ocean boundary is computed in Sv (1 Sv = 10$^6$ m$^3$ s$^{-1}$) from:

$$F_{w,lat} = \int_A v_t\, dA \qquad\qquad (Eq.\,6)$$

240 with $v_t$ the current velocity normal to the transect and $A$ the area of the section from the surface to bottom.

The lateral heat flux $F_{T,lat}$ in PW (PW $=10^{15}$ W) is computed from:

$$F_{T,lat} = \rho_0 C_p \int_A T v_t\, dA \qquad\qquad (Eq.\,7)$$

The lateral salt flux $F_{S,lat}$ in Gg s$^{-1}$ is computed from:

$$F_{S,lat} = \rho_0 \int_A S v_t\, dA \qquad\qquad (Eq.\,8)$$

245 Inflowing and outflowing fluxes are also computed using the same equations, but for values of $v_t >0$ and $v_t <0$, respectively:

$$F_{w,lat+} = \int_A v_t\, /\, (v_t > 0)\, dA \ \text{ and } F_{w,lat-} = \int_A v_t\, /\, (v_t < 0)\, dA \qquad\qquad (Eq.\,6')$$

$$F_{T,lat+} = \rho_0 C_p \int_A T v_t\, /(v_t > 0)\, dA \ \text{ and } F_{T,lat-}\rho_0 C_p \int_A T v_t\, /(v_t < 0)\, dA \qquad (Eq.\,7')$$

$$F_{S,lat+} = \rho_0 \int_A S v_t\, /\, (v_t > 0)\, dA \text{ and } F_{S,lat-} = \rho_0 \int_A S v_t\, /\, (v_t < 0)\, dA \qquad (Eq.\,8')$$

## 250 2.2.3 River fluxes

The river water volume flux $F_{w,riv}$ is calculated as the sum over all the rivers of the product of the velocity of river flow at the river mouth, $v_{riv}$:

$$F_{w,riv} = \sum_{rivers} \int_A v_{riv}\, dA \qquad\qquad (Eq.\,9)$$

where A is the area of the river mouth section from the surface to the bottom.

255 The river heat flux $F_{Triv}$, in PW, is computed from:

$$F_{T,riv} = \sum_{rivers} \rho_0 C_p \int_A T\, v_{riv}\, dA \qquad (Eq.\,10)$$

where T is the temperature (in °C) at the river mouth.

## 2.2.4 Atmospheric (surface) fluxes

The atmospheric freshwater volume flux is computed in Sv (1 Sv $= 10^6$ m$^3$ s$^{-1}$) from:

$$F_{w,surf} = \int_{Surf} (P - E)dxdy \qquad (Eq.\,11)$$

where P stands for the precipitation in m s⁻¹, E the evaporation in m s⁻¹, Surf is the SCS area limited by the six interocean straits shown in Fig. 1a.

The net surface heat flux ($F_{T,surf}$), in PW, is the sum over the SCS of the short-wave radiation flux ($F_{SR}$), long-wave radiation flux ($F_{LR}$), sensible heat flux ($F_{SEN}$) and latent heat flux ($F_{LATENT}$):

$$F_{T,surf} = \int_{Surf} (F_{SR} + F_{LR} + F_{SEN} + F_{LATENT})dxdy \qquad (Eq.\,12)$$

Finally, it should be noted that the flux calculations are numerically consistent with those carried out by the model through the advection scheme and its surface and continental boundary conditions. Along these lines, $C_p$ and $\rho_0$ constants correspond to the values used by the bulk formulas and the horizontal fluxes are calculated in the same way as in the advection scheme of the model. This allows to produce a strictly closed budget: the sum of all fluxes explains

100% of the variations of the water volume and of the total heat and salt contents at each time step of the simulation, as will be shown in Section 3.

### 2.3 Observational datasets

Satellite and tide gauges data are used for evaluating the representation of ocean surface characteristics (temperature, salinity, elevation). In-situ data are used to evaluate the surface and vertical representation of water mass properties

and the mixed layer depth (MLD).

### 2.3.1 Satellite data

To evaluate the modeled SST (sea surface temperature), we use daily OSTIA (Operational Sea Surface Temperature and Sea Ice Analysis) outputs for the period 2010 – 2018, available at https://data.nodc.noaa.gov/ghrsst/L4/GLOB/UKMO/OSTIA/ (last access 18 May 2023). OSTIA is a GHRSST (Group

for High Resolution Sea Surface Temperature) Level 4 SST daily product built from multiple spatial sensors and drifting and moored buoys data, with a horizontal resolution of 1/18°.

Regarding the SSS (sea surface salinity), we use outputs from the 9-day-averaged de-biased SMOS (Soil Moisture and Ocean Salinity) SSS Level 3 version 3, developed by Boutin et al. (2016). It has a resolution of 25 km and is available for the period 2010 – 2017. Data are distributed by the CECOS (Ocean Salinity Expertise Center) and the

CNES - IFREMER CATDS (Centre Aval de Traitement des Données SMOS) via:

https://data.catds.fr/cecos-locean/Ocean_products/L3_DEBIAS_LOCEAN/ (last access 18 May 2023).

To evaluate the SLA (sea level anomaly) and surface geostrophic currents, we use daily 1/4° global ocean gridded L4 sea surface heights in delayed – time of CMEMS, available at:

https://data.marine.copernicus.eu/product/SEALEVEL_GLO_PHY_L4_MY_008_047/description (last access 18 May 2023). This altimetry product (hereafter called ALTI) is generated using data from different altimeter missions and covers the period from 1993 up to present (Ablain et al. 2015; Ray and Zaron, 2016). For model-data comparison, we extracted the daily altimetric SLA on the period of comparison and removed at each point of each dataset (model and altimetry) the temporal average over the same period.

### 2.3.2 In-situ data

More than 12000 Argo profiles were collected in the SCS between 2010 and 2018 (see Fig. 1b), available from https://data-argo.ifremer.fr/geo/pacific_ocean/ (http://doi.org/10.17882/42182, last access 18 May 2023).

The ALIS R/V crossed the SCS from 10 May to 28 July 2014 (see Fig. 1b), measuring SST and SSS every 6 s by the vessel-mounted Seabird SBE21 thermosalinometer (hereafter called TSG-Alis data).

Under the framework of a cooperative Vietnam - US international research program (Rogowski et al. 2019), a Seaglider sg206 was deployed on 22 January 2017 until 16 May 2017 in the SCS (see Fig. 1b). It collected 555 vertical profiles of conductivity, temperature and pressure from an unpumped Sea-Bird Electronics CTD (SBE 41CP). Conductivity, temperature and depth were sampled at 5 s intervals in the upper 150 m, corresponding to a resolution finer than 1 m, and between 55 – 100 s below. All sensors were factory calibrated. Salinity was corrected for the thermal lag error using a variable flow rate (Garau et al. 2011).

Argo, TS-Alis and glider in-situ measurements are compared in Section 4 with modeled profiles at the nearest point (in position and time).

The third version of GESLA (Global Extreme Sea Level Analysis) dataset, released in 2021, consists of 5119 tidal records obtained from multiple sources around the world (Haigh et al. 2023). This quasi-global, higher-frequency tide gauges dataset can be obtained from https://www.gesla.org (last access September 2023). Tide gauges records from 46 stations are collected over the SCS region, then compared with modeled tidal outputs at the nearest point.

### 2.4. Other global and regional models

Besides the observational dataset, four widely used model outputs (COPERNICUS, INDESO, OFES and GLORYS, see Table 1) are collected and compared with our SYM4 simulation over the same geographic zone (0.6-24°N, 99-124°E) from 2010 to 2016, the common simulation period of all models. In addition, a SYMPHONIE simulation using exactly the same configuration as SYM4 but with a coarser horizontal resolution (12 km ~ 1/10°), referred to as SYM12 in the following, is performed over the same period to study the influence of horizontal resolution on the model performance.

**Table 1: Global and regional models used for the comparison with SYMPHONIE outputs.**

| Model | SYMPHONIE | Global Ocean Physics Analysis and Forecast, called COPERNICUS here | CMEMS global ocean eddy-resolving reanalysis GLORYS12v1, called GLORYS here | Infrastructure Development of Space Oceanography, called INDESO here | OFES (OGCM for the Earth Simulator) simulation from JAMSTEC (Japan Agency for Marine-Earth Science and Technology) ver 2., called OFES here |
|---|---|---|---|---|---|
| Periods of simulation | 2010-2018 | 1993-now | 1993 - 2020 | 2009-2016 | 1958-2016 |
| Spatial resolution | SYM4 : 4 km ~ 1/28° SYM12: 12 km ~ 1/10° | 1/12° | 1/12° | 1/12° | 1/10° |
| Number of vertical layers | 50 | 50 | 50 | 50 | 105 |
| Simulation zone | Regional 99°E - 124°E, 0.6°S - 24°N | Global | Global | Regional 90°E - 144°E, 20°S - 25°N | quasi-global |
| Assimilation | No | Yes | Yes | No | No |
| Tide included | Yes | No | No | Yes | Yes |
| Atmospheric forcing | ECMWF analysis 1/8° 3 hourly | ECMWF analysis 1/8° 3 hourly | ECMWF (ERA-Interim) 80km | ECMWF analysis 1/8° 3 hourly | JRA55-do ver.08 55km 3hourly |
| References | | https://data.marine.copernicus.eu/product/GLOBAL_ANALYSISFORECAST_PHY_001_024/description, last access August 2023 https://doi.org/10.48670/moi-00016 | https://data.marine.copernicus.eu/product/GLOBAL_MULTIYEAR_PHY_001_030/description, last access August 2023 https://doi.org/10.48670/moi-00021, | Tranchant et al. 2016 | Sasaki et al. 2020 https://www.jamstec.go.jp/ofes/ofes2.html, last access August 2023 https://doi.org/10.17596/0002029 |

**2.5. Statistical evaluation**

The simulated dataset S and observational dataset O (of the same size N) are compared using three statistical parameters: the bias, the Normalized Root Mean Square Error (NRMSE) and the Pearson correlation coefficient R:

$$Bias = \bar{S} - \bar{O} \qquad (Eq. 13)$$

$$NRMSE = \frac{\sqrt{\frac{1}{N}\sum_{i=1}^{n}(Si-Oi)^2}}{(O_{max}-O_{min})} \qquad (Eq. 14)$$

$$R = \frac{\sum_{i=1}^{N}(O_i-\bar{O})(S_i-\bar{S})}{\sum_{i=1}^{N}(O_i-\bar{O})^2 \sum_{i=1}^{N}(S_i-\bar{S})^2} \qquad (Eq. 15)$$

Where Si and Oi are respectively the simulated and observed series, and $\bar{S}$ and $\bar{O}$ their mean values. In Section 3, we use the same statistical evaluation methods for the comparison between online and offline computation of lateral oceanic fluxes: $S_i$, $O_i$, $\bar{S}$ and $\bar{O}$ are respectively replaced by $OF_i$ (the offline fluxes series), $ON_i$ (the online fluxes series), $\overline{OF}$ and $\overline{ON}$ (the corresponding mean values).

**3. Added-value of the online budget computation**

Computing online all the terms of the budget allows to calculate the exact net lateral fluxes through each lateral ocean boundary, hence to rigorously close the budgets at all time scales, but also to calculate the exact outflowing/inflowing fluxes at each time step. Computing the lateral term offline, using the modeled velocity, temperature and salinity at the output frequency, indeed relies on the assumption that the integral over the output period of the product of velocity and temperature (or salinity) is equal to the product of their integrals, thus that the turbulent term $\underline{u'T'}$ in the following

equation is negligible:

$$\overline{uT} = \overline{(\bar{u}+u')(\bar{T}+T')} = \overline{\bar{u}\bar{T}} + \overline{u'\bar{T}} + \overline{T'\bar{u}} + \overline{u'T'} = \bar{u}\bar{T} + \overline{u'T'} \qquad (Eq. 16)$$

where u is the velocity normal to the vertical section and T the temperature at this point, and the overbar stands for the integral over the output period.

Here we quantitatively show the added value of the online computation of water volume, heat and salt budgets

compared to the offline computation. In Fig. 2a,b,c we show each term of the budget equation for the interannual variations of water volume, heat and salt contents over the SCS, computed online in SYM4 and offline from SYM4 outputs : annual variation, atmospheric surface fluxes, river fluxes, lateral oceanic fluxes and the sum of all fluxes, that should equal the annual variation as explained in Section 2.2. Table 2 provides the values of net, inflowing and outflowing annual fluxes computed online, and the bias, correlation and NRMSE between the offline and online

computations.

First, those figures confirm that when computed online, the sum of annual fluxes is equal to the annual variation, i.e., that the budget equation is closed in our model. This is shown here for the interannual variations but is also verified

at each time step of the whole simulation (figure not shown). Second, Fig. 2a,b,c quantitatively highlights the error induced when neglecting the turbulent term in Eq. 16 by computing the lateral net fluxes offline. For the water volume flux, using the online (blue) and offline (cyan) computation for net lateral oceanic fluxes is equivalent since it does not imply any non-linear assumption. For the heat and salt fluxes however, the difference is significant: we obtain NRMSE of, respectively, 34% and 8% between the online and offline computations for, respectively, heat and salt net lateral fluxes over the SCS for the 2010-2018 period (Table 2).

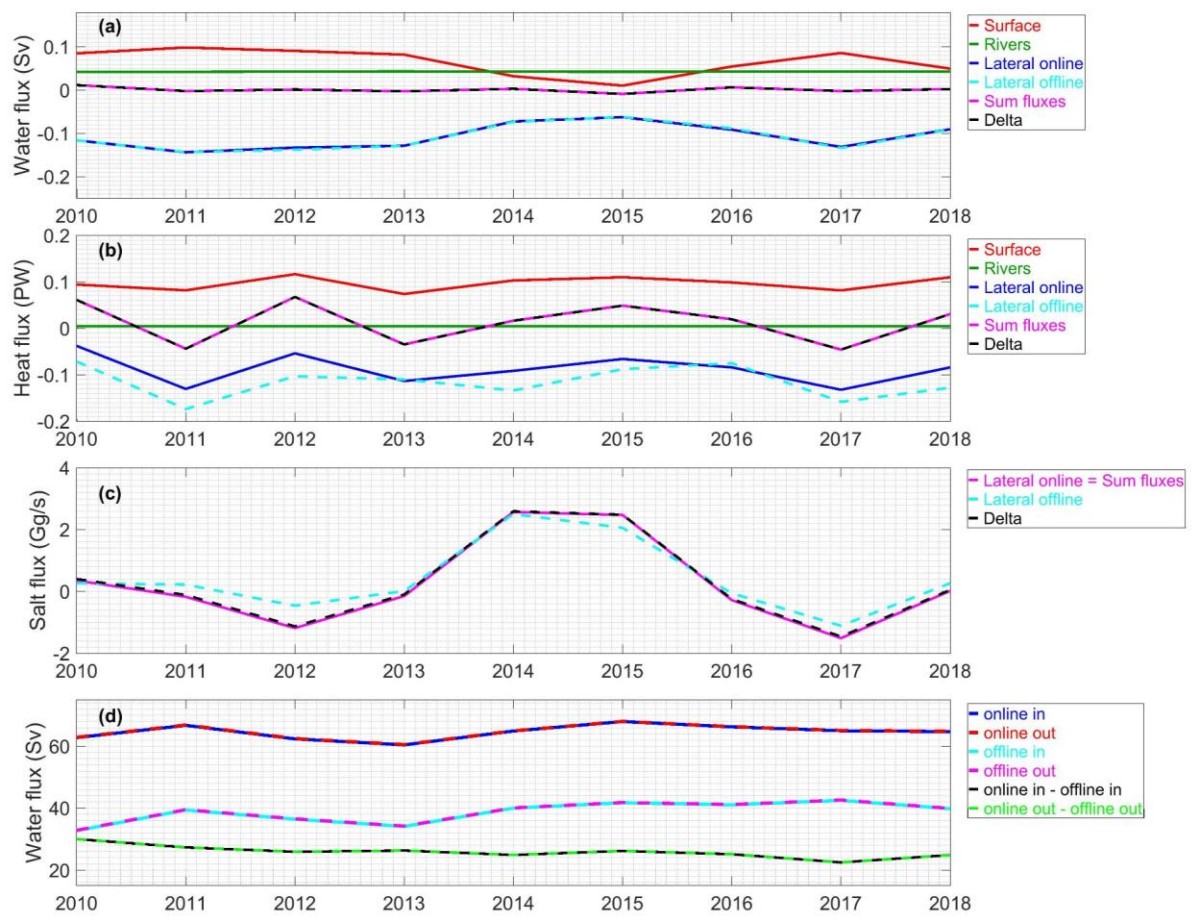

**Figure 2 : Atmospheric (surface, red), river (green) and net lateral oceanic (blue online, cyan offline) annual fluxes of (a) water volume (Sv), (b) heat (PW) and (c) salt (Gg.s⁻¹) and their sum (magenta), and annual variations of water volume, heat and salt contents (black) over the period 2010 - 2018; (d) Annual lateral oceanic inflow (blue online, cyan offline) and outflow (red online, magenta offline) of water volumes (in absolute values) computed online and offline, and the difference online - offline (black for inflow, green for outflow).**

Third, the online computation allows us to compute separately the outflowing and inflowing terms of the lateral flux at each time step. Fig. 2d shows the annual water volume lateral inflowing and outflowing fluxes (in absolute values) computed online and offline. Using the offline computational methods leads to important errors: the offline

computation underestimates the water volume outflow and inflow by a factor of ~2 (Table 2). Correlations of online and offline water volume, heat and salt annual inflows or outflows are statistically significant (~ 0.80, p-value <0.01), showing a similar chronology in both methods. However, the bias between online and offline inflowing or outflowing lateral water volume, heat and salt fluxes is ~40% compared to the mean value, and high NRMSE values (~330%, 210% and 315% for water volume, heat and salt respectively, Table 2) are obtained. These results quantitatively demonstrate the significant errors made when computing those fluxes offline and show the relevance of the online computation.

**Table 2: Mean values over 2010-2018 of water volume, heat and salt net, inflowing and outflowing annual fluxes through the SCS computed online (1st column), and absolute (2nd) and relative (3rd) bias, correlation (4th) and NRMSE (5th) between online and offline computations.**

| Lateral flux | Mean value in online computation | Bias (offline - online) | Bias/mean (relative bias) in % | Correlation (offline/online) | NRMSE in % |
|---|---|---|---|---|---|
| Water volume net (Sv) | -0.108 | -5E-4 | 0.46 | 1.00 (p=0.00) | 2.44 |
| Heat net (PW) | -0.088 | -0.028 | 31.3 | 0.81(p<0.01) | 34.1 |
| Salt net (Gg/s) | 0.236 | 0.176 | 74.6 | 0.99 (p=0.00) | 8.35 |
| Water volume in (Sv) | 64.6 | -25.9 | 40.1 | 0.81 (p<0.01) | 327 |
| Water volume out (Sv) | -64.7 | 25.9 | 40.1 | 0.81 (p=0.01) | 330 |
| Heat in (PW) | 3.34 | -1.29 | 38.6 | 0.82 (p<0.01) | 205 |
| Heat out (PW) | -3.42 | 1.23 | 36.0 | 0.85 (p<0.01) | 213 |
| Salt in (Gg/s) | 2285 | -914 | 40.0 | 0.82 (p<0.01) | 313 |
| Salt out (Gg/s) | -2285 | 914 | 40.0 | 0.82 (p<0.01) | 316 |

**4. Model performance in representing sea surface and water masses characteristics**

In this section, we evaluate the ability of SYM4 to simulate over 2010-2018 the characteristics of SCS sea surface (temperature, salinity, sea surface elevation including tides), water masses and mixed layer depth.

**4.1 Sea surface characteristics**

We evaluate here the ability of SYM4 to represent sea surface characteristics and their variability at the tidal, seasonal and interannual scale by comparing them with tide gauges data, tide reanalysis and satellite observations of sea surface temperature, salinity and elevation.

**4.1.1 Tides**

The tide representation over the coastal zone is evaluated by comparing SYM4 results with the 46 GESLA tide gauges data available over the SCS. Results are presented in Fig. 3. We obtain similar simulated and observed values in amplitudes and phases with relatively weak biases for the four main tidal components (K1, O1, M2 and S2). The SCS is indeed one of the few regions of the global ocean where diurnal tides (K1, O1) dominate semi-diurnal tides (M2, 385 S2) (Guohong, 1986). SYM4 overestimates diurnal tides amplitude by ~ 15 cm in the southern Gulf of Thailand for K1 and underestimates it by ~10 cm in the Gulf of Tonkin and Sulu Sea coastal zone for O1 (Fig. 3c1,c2). Concerning the semi-diurnal tide, amplitude differences are about ± 5 cm for most of stations, the strongest biases are observed at the Sulu Sea (~20 cm for M2 and ~10 cm for S2) and Celebes Sea (~ -20 cm for M2 and S2) (Fig. 3c3,c4).

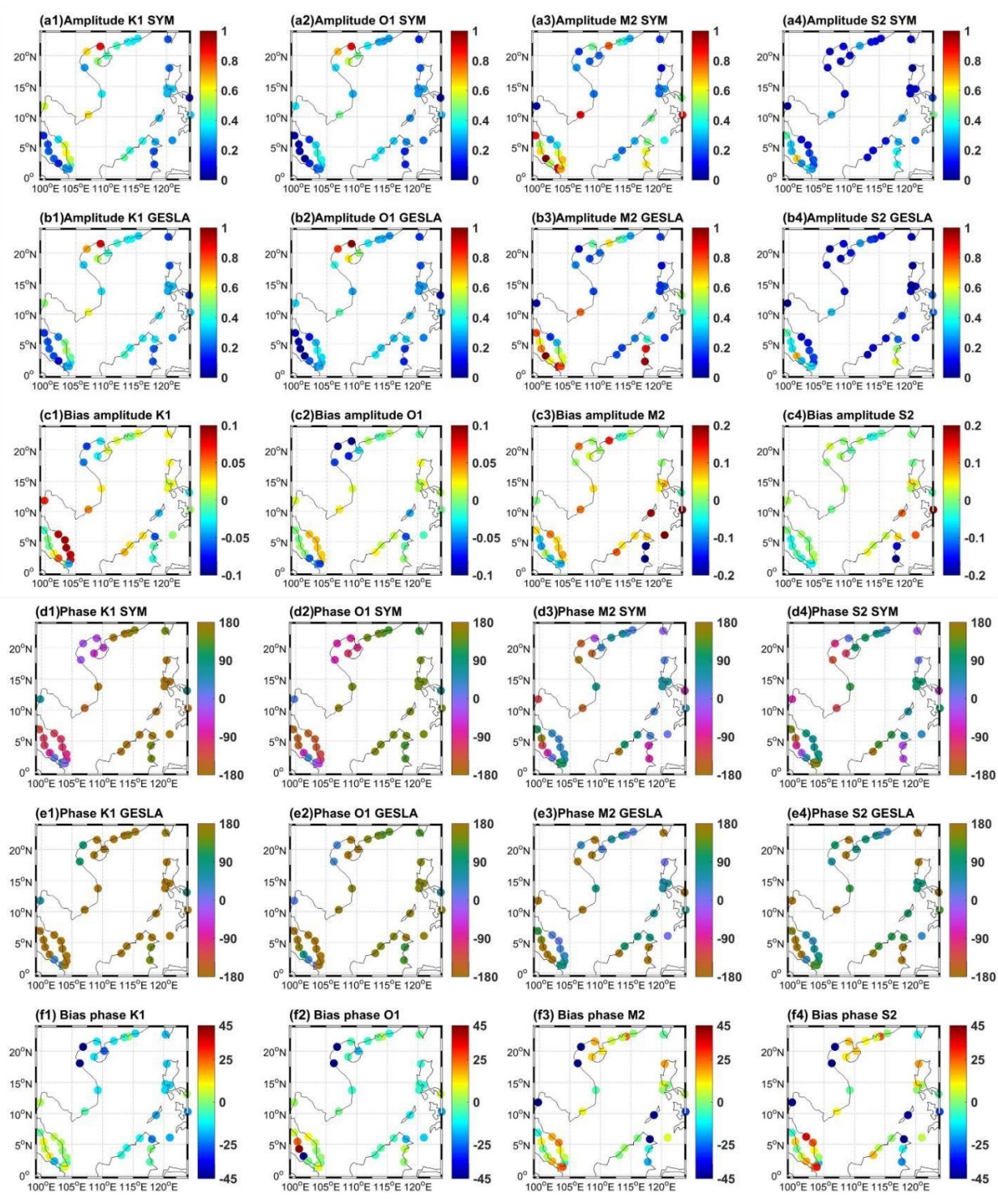

Figure 3 : Amplitude (m) and phase (degree) of four tidal components K1, O1, M2, S2 in SYM4 (SYM, a1-a4, d1-d4) and GESLA tide gauge dataset (b1-b4, e1-e4) and model bias compared to GESLA (c1-c4, f1-f4)

FES2014b is used to provide the tidal forcing at the lateral boundaries of our numerical domain, located outside the SCS (Fig. 1a). FES2014b assimilates satellite and in-situ sea surface elevation data, allowing it to be very close to observations, as shown by Piton et al. (2020) over the Gulf of Tonkin. We therefore also use it, complementary to GESLA tide gauges data, to evaluate the tidal solution produced by our model over the inner open sea domain. We show in Fig. 4 the observed and simulated tidal amplitude and phase for K1, O1, M2 and S2, the four principal tidal components in the SCS region. The spatial distribution of tidal constituents obtained from SYM4 and from FES2014b is similar to the study of Phan et al. (2019). Diurnal tides prevail over the Gulf of Tonkin, the Gulf of Thailand and the southwestern SCS. Mixed tides (mainly semi-diurnal tides) prevail along southern China, the northwest coast of Borneo and the continental shelf of the Mekong delta. For those four tidal components, we obtain a strong similarity both for amplitudes and phases between SYM4 and FES2014b over most of the modeled domain. As observed from the comparison with tide gauges, the most noticeable weaknesses are a small (< 10 cm) underestimation of diurnal (K1 and O1) amplitude in the Sulu Sea and Gulf of Tonkin, an overestimation (~20 cm) of semi-diurnal (M2 and S2) amplitude in the Sulu Sea, and a small overestimation of K1 amplitude off the Mekong Delta. The bias of semi-diurnal tidal amplitudes in the Sulu Sea may be related to the prescribed bathymetry in the area, with many small islands separating this area from the surrounding seas.

Comparing tidal amplitude biases in SYM4 and in SYM12 shows that the high resolution significantly reduces the biases over the whole domain (see Fig. A3): the strong O1 and K1 biases in the Sulu Sea in SYM12 are reduced by ~80-90% in SYM4, and the M2 and S2 biases near the southern China and Vietnam coasts and western Borneo coast by ~20-30%. This improvement of tidal solution in SYM4 can be partly attributed to the better representation of bathymetry details, in particular in the interocean straits (see Figs. A1, A2). Comparing SYM4 results with FES2014b tidal solution, tide gauge data and with SYM12 therefore shows that SYM4 reproduces realistically the tidal solution in the SCS, both in the open sea and in the coastal area, and that the high resolution helps to improve the realism of this tidal solution.

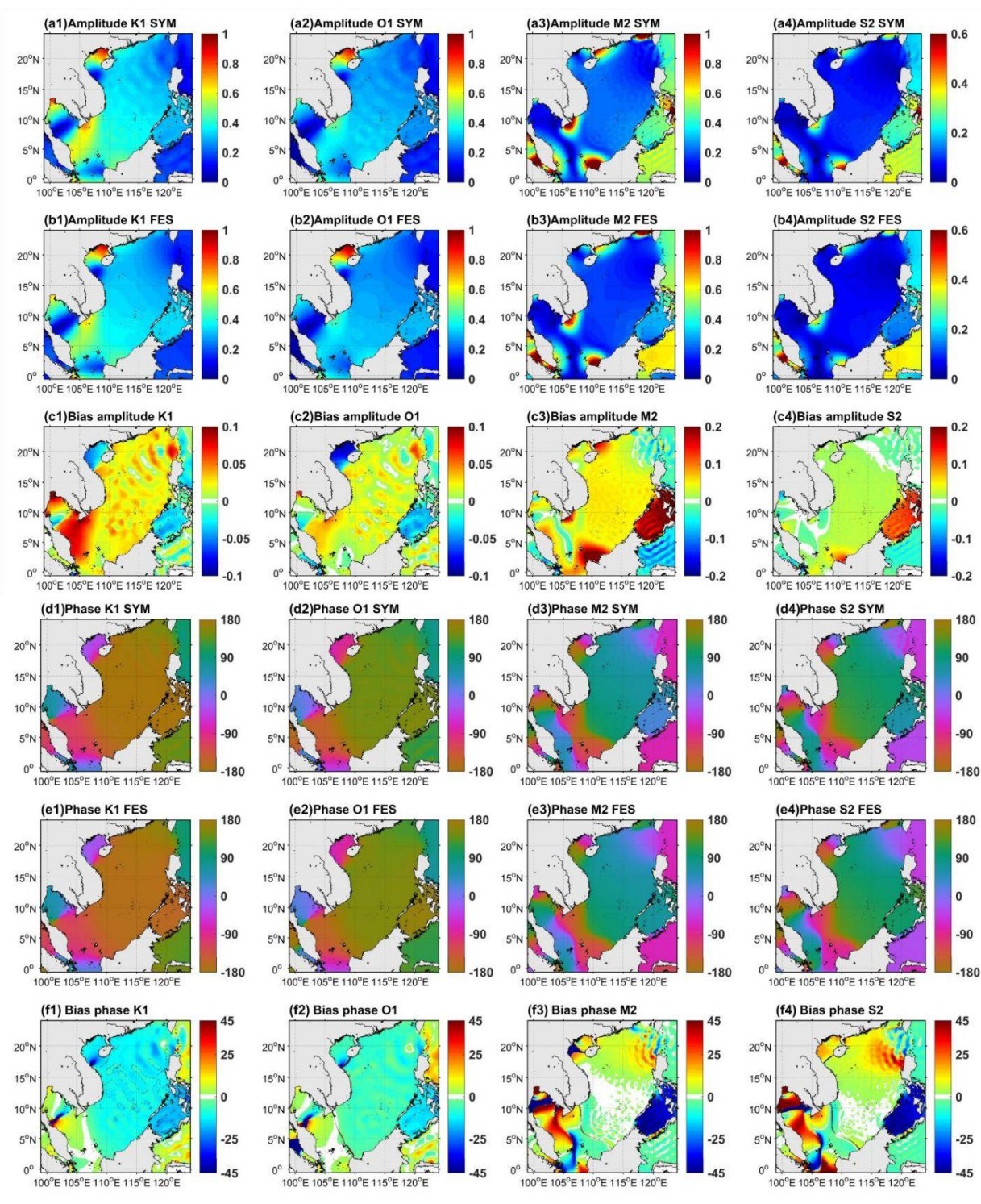

**Figure 4: Amplitude (m) and phase (degree) of four tidal components K1, O1, M2, S2 in SYM4 (SYM, a1-a4, d1-d4) and the global tidal product FES2014b (FES, b1-b4, e1-e4) and the bias in SYM4 compared to FES2014b (c1-c4, f1-f4).**

**4.1.2 Seasonal cycle of sea surface temperature, salinity and elevation**

We show in Fig. 5 the seasonal cycle (Fig. 5a,c,e) and the interannual variations (Fig. 5b,d,f) of SST, SSS and SLA
computed from model outputs (for SYMPHONIE as well as COPERNICUS, INDESO, GLORYS and OFES) and
from the corresponding satellite observations. Table 3 shows the corresponding bias, NRMSE and correlation
coefficients.

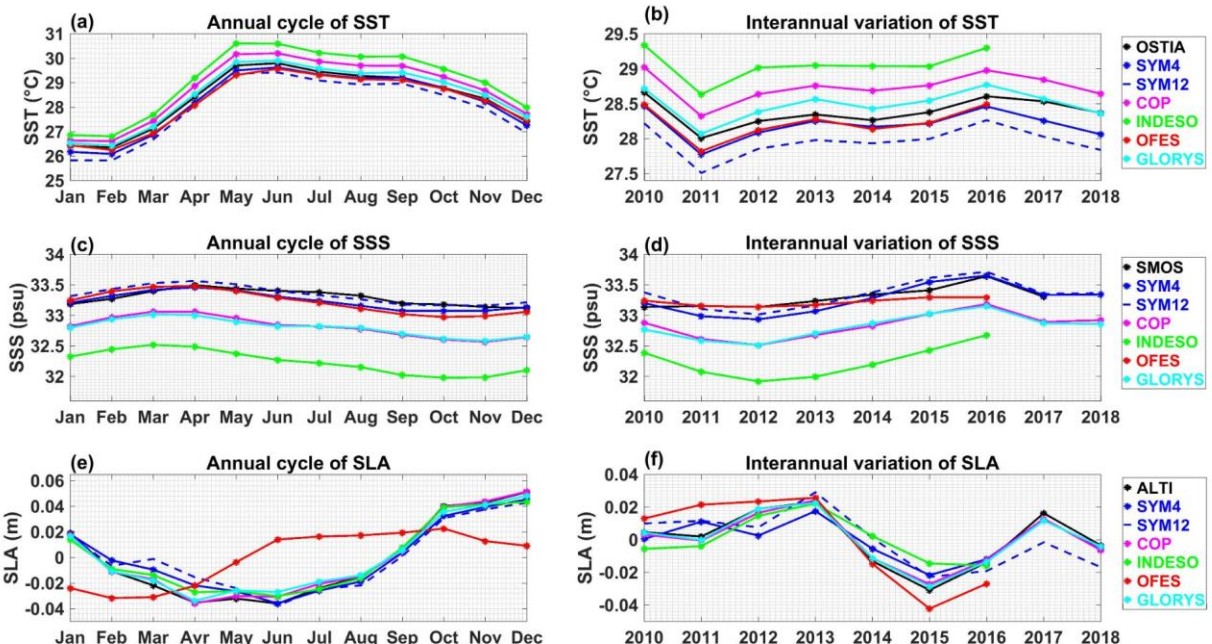

**Figure 5 : Time series of climatological monthly mean (left, computed over 2010-2016, the period common to all models)**
**and yearly mean (right, over 2010-2018 except for OFES and INDESO, available only until 2016) of SST (°C, a, b), SSS**
**(psu, c, d) and SLA (m, e, f) averaged over the SCS domain, computed from different models (SYM4, blue; SYM12 blue**
**dashed line, COPERNICUS, magenta; INDESO, green; OFES, red; GLORYS, cyan) and from satellite observations**
**(OSTIA, SMOS and ALTI, black).**

The annual cycle of SST averaged over the SCS (Fig. 5a) is very well simulated in SYM4, with a highly significant
correlation (R=0.99 and p-value p < 0.01, corresponding to a significant level higher than 99%) and a small NRMSE
(5.7%) between SYM4 outputs and OSTIA, and a slight bias of -0.18 °C for the period 2010 - 2018. In all datasets,
the monthly climatological cycle of SST reaches its maximum value in May/June (spring-summer) and decreases to
its minimum in January/February (winter). This monthly climatological SST agrees with the study of Kumar et al.
(2010), who observed the same SST annual cycle by analyzing hydrographic WOA data (World Ocean Atlas, 2005).

SYM4 SSS seasonal cycle (Fig. 5c) also shows a good agreement with SMOS data, with a highly significant
correlation of 0.91 (p <0.01), a low NRMSE equal to 19%, and a slight negative bias (-0.04 psu). In both model and
data, the average SSS is maximum in April (spring) with values of 33.47 psu in SYM4 and 33.52 psu in SMOS, and

minimum from September to December (autumn) with 33.07 psu in SYM4 and 33.17 psu in SMOS. This significant seasonal variation of SSS in the SCS, with high salinity in winter-spring and low salinity in summer-autumn was also obtained by Kumar et al. (2010) and Zeng et al. (2014).

The annual cycle of SLA obtained with SYM4 and ALTI data during the period 2010 - 2018 shows a minimum value in spring-summer (June) with -0.033 m both for SYM4 and ALTI (Fig. 5e). The SLA reaches its highest value in winter (December) with 0.039 m and 0.049 m respectively for SYM4 and ALTI. SYM4 outputs and the altimeter measurements have a highly significant correlation (R=0.97, p<0.01), and a small NRMSE value (10%). The simulated monthly climatological SLA is also in agreement with Shaw et al. (1999) and Ho et al. (2000): using TOPEX/Poseidon altimeter data, they both concluded on a higher SLA in winter and lower SLA in summer over the SCS.

We then show the simulated and observed maps of SST, SSS and SLA climatologically averaged over the boreal winter (December, January and February, DJF) and summer (June, July and August, JJA) for SYM4 (Fig. 6) as well as the corresponding bias for the 6 simulations (Fig. 7).

In both winter and summer, SYM4 SST is very close to observations, with highly significant spatial correlation (respectively R=0.99 and 0.84 in winter and summer, p<0.01) and similar ranges compared to OSTIA (Fig. 6a,b). In winter, SYM4 shows an average negative bias of -0.28 °C, and colder zones offshore southern Vietnam and in the northern basin. In summer (Fig. 6c,d), the average negative bias is reduced to -0.17 °C, and the simulation produces a SST colder than OSTIA in the northern SCS near Taiwan, off southern Vietnam coast, along the Mekong delta, and in the Sulu and Celebes seas (see Fig. 1a). On the other hand, simulated SST is warmer in the Gulf of Tonkin, Gulf of Thailand and the southern basin.

SYM4 spatial distribution of SSS also shows a highly significant spatial correlation with SMOS for both seasons (R=0.88 and 0.84 psu in winter and summer, respectively, p<0.01). SYM4 has a positive bias in winter (0.05 psu), and a negative bias in summer (-0.1 psu). In winter (Fig. 6e,f), the Chinese and Vietnam coastal zones and the Gulf of Thailand are fresher in SYM4 than in SMOS data, whereas the center of the basin and the southern Gulf of Tonkin are saltier. In summer (Fig. 6g,h), we obtain a significantly lower SSS at the big river mouths (Pearl River, Red River, Mekong River), in the Gulf of Thailand and in the Malacca Strait in model outputs compared to SMOS. SMOS, with a resolution of 25 km, might however not be able to capture these salinity changes in the coastal zone.

Both in winter and summer, the simulated and observed seasonal mean spatial distributions of SLA show a highly significant correlation (R=0.97, p<0.01, Fig. 6i,j,k,l). SYM4 shows very weak negative seasonal biases in SLA compared to ALTI (-0.008 m in winter and -0.003 m in summer). In the Gulf of Thailand, the simulated SLA is lower in winter and higher in summer compared to ALTI. Regarding the geostrophic currents, we obtain great similarities between the model and ALTI. In winter when the northeastern monsoon dominates, two cells of cyclonic gyre cover the whole basin, one near Luzon and another at the Sunda shelf. In summer, with the southwest monsoon, most of the SCS geostrophic currents reverse and flow northeast. The geostrophic currents are most intense at the Sunda shelf zone (see Fig. 1a) in winter. In summer, we observe strong geostrophic flows at the southern Vietnam coast, and at

the east of the Malaysian coast. The intensity and direction of those seasonal geostrophic currents are consistent with previous studies (e.g., Da et al. 2019; Wang et al. 2006b).

Last, Fig. 8i,j shows the observed TSG-Alis SST and SSS during spring-summer 2014 and the corresponding colocalized simulated SSS and SST. Again, SYM4 shows a strong similarity with TSG-Alis data, with correlation coefficients of 0.70 and 0.82 (p<0.01), for SST and SSS respectively, during this 6th year of the simulation.

**Table 3: Bias, correlation coefficients and NRMSE values (for the climatological monthly annual cycle and interannual yearly time series (in italics) shown in Fig. 5) for the SST, SSS and SLA simulated by SYMPHONIE, OFES, INDESO, COPERNICUS and GLORYS compared to satellite observations (OSTIA, SMOS and ALTI, respectively), and for the climatological monthly annual cycle of simulated MLD shown in Fig. 9 compared to Argo data. The period over which indicators are computed is indicated below the model name**

| Models and period | | SYM4 2010-2018 4km ~1/28° | SYM4 2010-2016 4km ~1/28° | SYM12 2010-2016 12km ~ 1/10° | OFES 2010-2016 | INDESO 2010-2016 | COPERNICUS 2010-2016 | GLORYS 2010-2016 |
|---|---|---|---|---|---|---|---|---|
| Bias | SST (°C) | -0.18 | -0.16 | -0.40 | -0.14 | 0.70 | 0.38 | 0.14 |
| | SSS (psu) | -0.04 | -0.05 | 0.04 | -0.07 | -1.05 | -0.48 | 0.14 |
| | SLA (m) | -4.5E-4 | 1.2E-3 | 1.2E-3 | 1.3E-3 | 1.2E-3 | 1.2E-3 | 1.3E-3 |
| | MLD (m) | 9.4 | 9.29 | 12.5 | 7.13 | 15.4 | 15.3 | 10.5 |
| Correlation coefficient R (%) annual cycle/ *interannual* | SST | 0.99 p<0.01 *0.94 p<0.01* | 0.99 p<0.01 *0.98 p<0.01* | 0.99 p<0.01 *0.94 p<0.01* | 0.99 p<0.01 *0.98 p<0.01* | 0.99 p<0.01 *0.97 p<0.01* | 0.99 p<0.01 *0.99 p<0.01* | 0.99 p<0.01 *0.97 p<0.01* |
| | SSS | 0.91 p<0.01 *0.91 p<0.01* | 0.86 p<0.01 *0.91 p<0.01* | 0.86 p<0.01 *0.86 p=0.01* | 0.80 p<0.01 *0.78 p=0.04* | 0.72 p<0.01 *0.78 p=0.04* | 0.83 p<0.01 *0.86 p=0.01* | 0.83 p<0.01 *0.92 p<0.01* |
| | SLA | 0.97 p<0.01 *0.90 p<0.01* | 0.98 p<0.01 *0.89 p<0.01* | 0.96 p<0.01 *0.89 p<0.01* | 0.31 p=0.3 *0.95 p<0.01* | 0.99 p<0.01 *0.85 p=0.01* | 0.99 p<0.01 *0.99 p<0.01* | 0.99 p<0.01 *0.99 p<0.01* |
| | MLD | 0.98 p<0.01 | 0.98 p<0.01 | 0.92 p<0.01 | 0.99 p<0.01 | 0.97 p<0.01 | 0.98 p<0.01 | 0.99 p<0.01 |
| NRMSE (%) annual cycle/ *interannual* | SST | 5.73 *26.2* | 5.15 *24.8* | 11.7 *74* | 5.40 *21.5* | 20.7 *107* | 11.2 *58.0* | 4.27 *22.8* |
| | SSS | 18.9 *20.0* | 23.4 *26.1* | 22.3 *26.7* | 35.4 *29.9* | 282 *211* | 129 *96.9* | 132 *98.4* |
| | SLA | 9.92 *17.5* | 7.85 *16.5* | 11.3 *15.5* | 35.9 *19.6* | 5.74 *18.1* | 2.77 *4.59* | 4.71 *3.45* |
| | MLD | 24.3 | 25.8 | 42.5 | 19.2 | 40.2 | 39.8 | 27.5 |

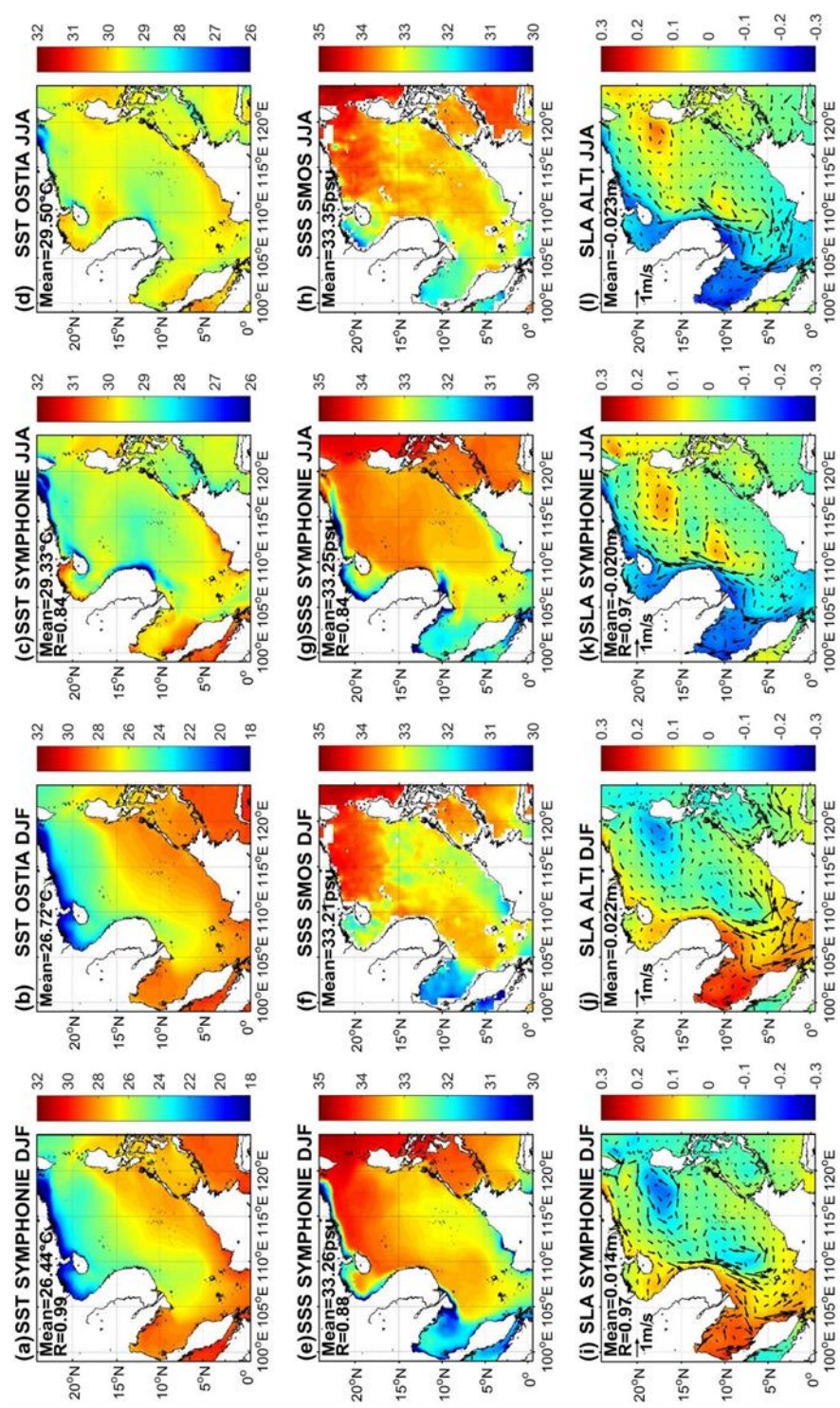

**Figure 6 : Spatial distribution of winter (DJF) and summer (JJA) climatologically averaged SST (°C, a, b, c, d), SSS (psu, e, f, g, h), SLA (m) and geostrophic current (m/s, i, j, k, l) in SYM4 outputs and corresponding satellite observations over 2010-2018. R stands for the spatial correlation coefficient (here p-value is always smaller than 0.01)**

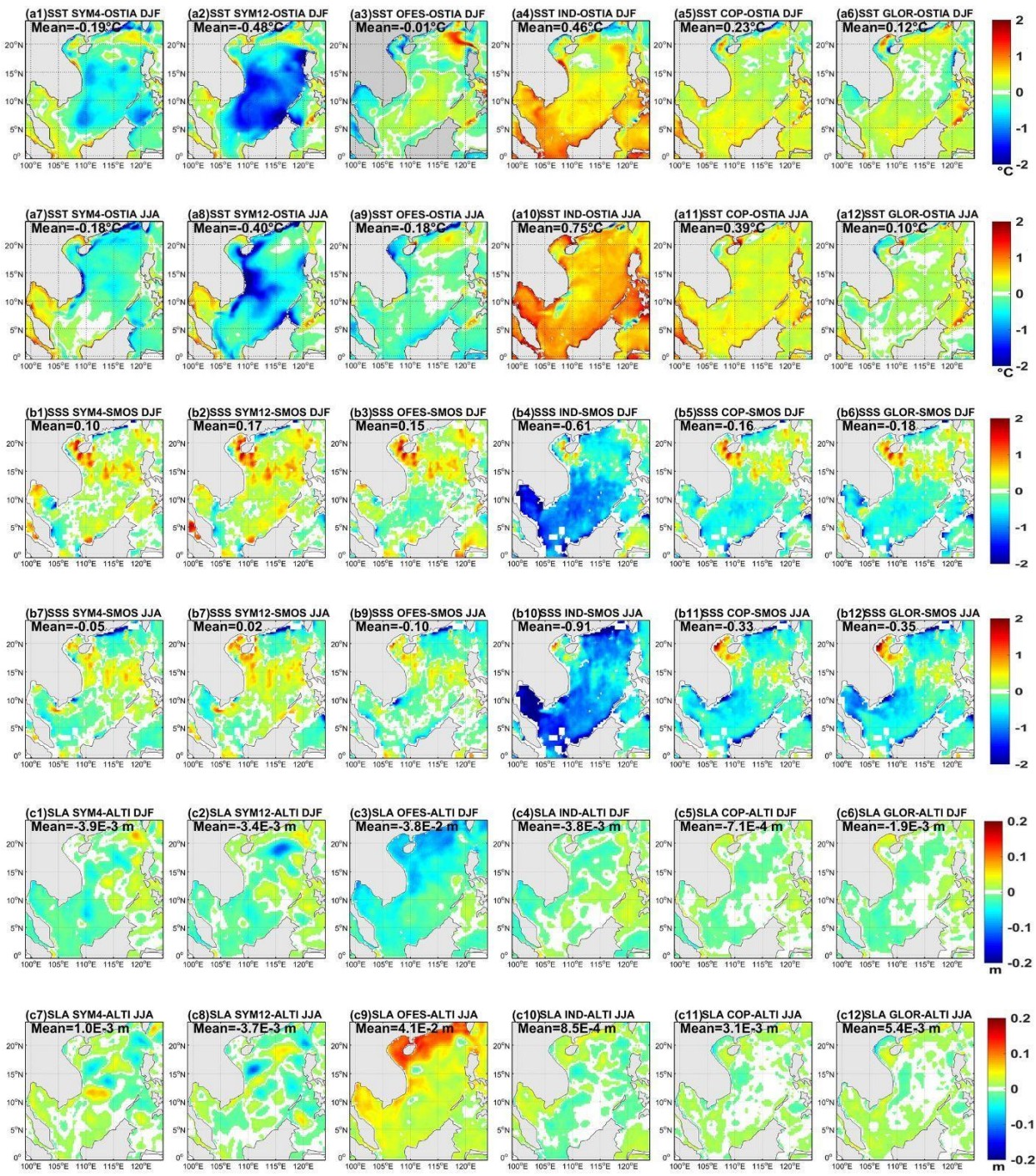

**Figure 7: Maps of biases between models and satellite datasets averaged in winter (December, January, February, DJF) and summer (June, July, August, JJA) during the period 2010 - 2016 for SST (°C, compared to OSTIA, a1-a12), SSS (psu, compared to SMOS, b1-b12) and SLA (m, compared to ALTI, c1-c12).**

### 4.1.3 Interannual variations of sea surface temperature, salinity and elevation

We obtain a highly significant correlation coefficient between SYM4 and OSTIA (R=0.94, p<0.01) regarding yearly SST interannual variations (Fig. 5b). The yearly -0.18 °C SST bias is nearly constant over the period and the NRMSE is 26%. From 2010 to 2018, the averaged yearly SST over the basin reaches its highest values in 2010 and 2016 (28.47°C and 28.46°C, respectively). This is consistent with the study of Yu et al. (2019), who found a co-occurrence between those SST positive anomalies peaks and El-Niño events in 2009-2010 and 2015-2016 (see the NOAA ONI time series available at https://origin.cpc.ncep.noaa.gov/products/analysis_monitoring/ensostuff/ONI_v5.php). The minimum of averaged SST (27.77 °C) occurs in 2011, corresponding to the 2011-2012 La Niña event. Yu et al. (2019) obtained the same interannual time-series by analyzing MODIS satellite-derived SST data for the period 2003 - 2017.

The simulated interannual variations of yearly SSS (Fig. 5d) show a highly significant correlation (R=0.91, p<0.01) and a rather low NRMSE value (20%) with satellite data. There is a significant increase of the annual averaged SSS over the SCS between 2012 to 2016, both in SYM4 outputs and SMOS data. Over the period 2010 - 2017, the SSS reaches a low value in 2012 (32.93 psu for SYM4, 33.14 psu for SMOS), then increases continuously until a maximum value in 2016 (33.65 psu for SYM4, 33.64 psu for SMOS). The freshening until 2012 and strong salinification during the following four years are in agreement with observations of Zeng et al. (2014, 2018), who revealed that 2012 was the year with the lowest recorded value of SSS in the SCS over a 50-year period, and that the SSS then increased from late 2012 to 2016.

In terms of SLA interannual variations, SYM4 and ALTI show strong similarities with a NRMSE equal to 17.5% (Fig. 5f) and a highly significant correlation (R=0.90, p<0.01). During the studied period, the overall averaged SLA is maximum in 2013 (0.017 m in model outputs and 0.023 m in ALTI), and minimum in 2015 (-0.02 m in SYM4 and -0.03 m in ALTI).

### 4.1.4 Comparison with other models

SYM4 performances in representing sea surface characteristics are also compared with other numerical dataset over 2010 - 2016, the period common to all simulations (Fig. 5 and 7 and Table 3). The most striking weaknesses are an underestimation of SST (-0.4 °C) in SYM12, an overestimation of SST (0.7 °C) and underestimation of SSS (-1.1 psu) in INDESO and a wrong representation of SLA seasonal cycle in OFES (correlation of 0.31). Apart from this, all models compare well with observations in terms of bias, seasonal cycle and interannual variability, and spatial variability. Moreover, SYM4 is always in the upper performance range, for bias, correlation coefficients and RMSE. In comparison with other models and with SYMPHONIE at 12 km resolution, SYM4 thus shows a good performance in simulating the seasonal cycle and interannual of surface characteristics, and performs as well or even better than models that include assimilation (GLORYS and COPERNICUS).

Those comparisons of SYM4 SST, SSS and SLA time series and spatial fields with observations dataset and other simulations available from models at coarser resolution, including SYMPHONIE, therefore shows the added-value of

our high-resolution simulation in realistically reproducing the annual cycle and interannual variations and the seasonal spatial distributions of SCS surface hydrological characteristics and circulation over the period 2010-2018.

**4.2 Water masses characteristics**

We hereafter examine the performance of SYM4 in simulating the vertical distribution of water masses temperature and salinity properties. For that, we compare model results with Argo and glider observations. Fig. 8a-h show the colocalized simulated and observed temperature and salinity profiles, their mean value and the bias between model and data.

We obtain a strong agreement between the simulated and observed temperature and salinity profiles both for Argo floats (over the period 2009-2018) and glider (winter-spring 2016) outputs (Fig. 8a-h). In particular the maximum salinity observed in the intermediate water mass, corresponding to the Maximum Salinity Water (MSW), is well reproduced by SYM4. In general, modeled temperatures are lower than measured temperatures, with a negative bias in the whole water column (Fig. 8b,f). The highest biases are located in the subsurface layer (50-200 m), with maximum biases of -1.2°C compared to Argo data and of -1.5°C compared to glider data. Under 200 m, the temperature bias is stable, varying around 0.2-0.5°C compared to Argo floats, and 0.7-1°C compared to glider data. Model results show a general very low positive salinity bias compared to Argo and glider data below 200 m. A higher salinity bias is obtained in the subsurface layer: 0.2 psu compared to Argo data (Fig. 8d) and 0.3 psu compared to glider data (Fig. 8h). Therefore, our simulation accurately represents the various SCS water masses characteristics over the water column. It is moreover noteworthy that in 2016, i.e. the 7th year of simulation, those characteristics are reproduced without a significant drift.

SYM4 also shows a good performance in reproducing the temperature and salinity vertical distribution in comparison with other numerical datasets. Temperature and salinity profiles averaged over the zone 112-118°E, 12-18°N and the period 2010-2016 are shown in Fig. 10a. COPERNICUS and GLORYS profiles are closest to Argo thanks to the assimilation. Concerning the models without assimilation, INDESO shows the lowest temperature bias (maximum 0.4°C at 100-150 m depth) compared to Argo. Higher temperature biases are observed with SYM4 (~ -1.5°C at 50-100m depth) and OFES (~ -0.7°C at 800m). The strongest biases in temperature profiles are obtained from SYM12, with negative bias of -3.5 °C at 50 m depth and a positive bias of 0.8 °C at 1000-1200 m depth. Regarding the salinity, SYM4 shows the lowest bias for the upper 0-300 m layer (~0-0.3 psu), while INDESO shows a strong surface fresh bias (up to 0.8 psu), OFES and SYM12 respectively overestimates and underestimates the salinity maximum (by ~0.5 psu at 100-200 m depth for OFES and by -0.17 psu for SYM12). Underneath 200 m, all models present low salinity biases in comparison with observation and OFES shows the strongest bias (~ -0.1 psu at 500 m).

Argo floats, glider and model data show water masses characteristics in agreement with previous studies done on water masses over the SCS (Uu and Brankart, 1997; Penjan et al. 1997; Rojana-anawat et al. 1998; Saadon et al. 1998a, b) and the Pacific (Talley et al. 2011) (Fig. **8**a-h). In the upper layer (0-50 m), we observe both the Open Sea

Water (OSW), characterized by salinities of 33-34 psu and temperatures of 25-30 °C, and the Continental Shelf Waters (CSW) with low salinities (< 33 psu) and temperatures between 20 and 30°C (depending on the season). The 50-100 m layer is characterized by the mixing between the Northern Open Sea Water (NOSW) and the Pacific Ocean Water

(POW) during winter. The NOSW has salinities of 34-34.5 psu and temperatures of 23-25 °C. The POW is saltier with salinities of 34-35 psu and temperatures of 25-27 °C. Deeper, at 100-200 m, the MSW is characterized by temperatures between 15-17 °C and salinities between 34.5 psu and 35 psu and is a property of the equatorial regions (Rojana-anawat et al. 1998). Below the MSW, from 200-1000 m, the North Pacific Intermediate Water (NPIW) and Pacific Equatorial Water (PEW) are flowing with temperatures and salinities between 5-13 °C and 34-35 psu,

respectively. The Deep Water (DW), below 1000 m, is identified by temperatures of 2-5 °C, and salinities of 34.3 - 34.7 psu. Temperature profiles located in the Sulu Sea do not follow those characteristics in the deep layers, both in Argo and model outputs, showing temperature varying from 7 to 10°C below 700 m. This marginal sea, nearly isothermal, indeed possesses unique water characteristics, with a potential temperature varying around 9.8°C below 1000 m (Wyrtki, 1961; Chen et al. 2006; Gordon et al. 2011), much higher than those of neighboring seas such as the

SCS, the Celebes Sea and the Western (Qu and Song, 2009).

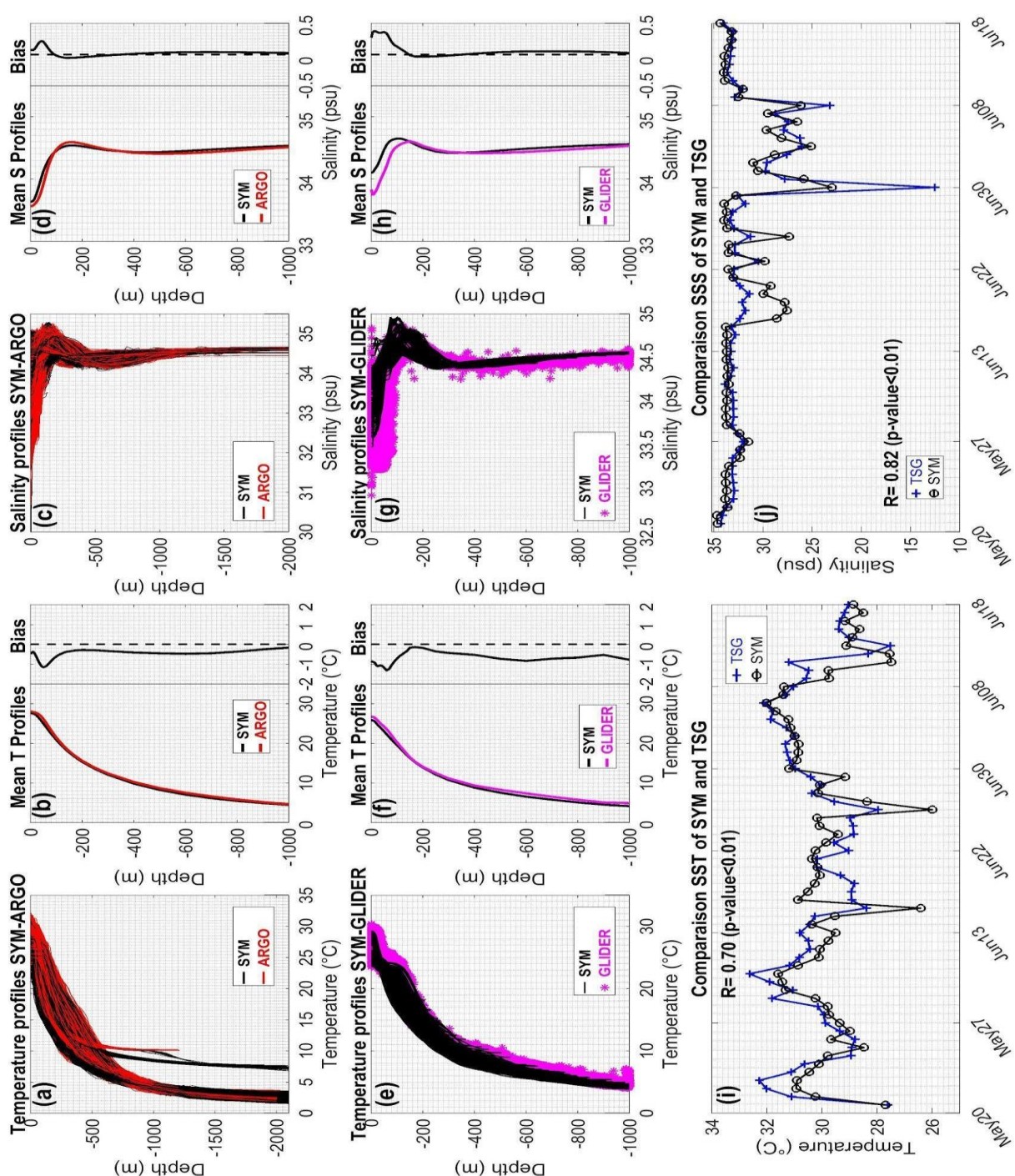

**Figure 8 : Temperature (°C) and salinity (psu) vertical profiles (all profiles, mean profiles and mean bias between model and observations) from SYM4 outputs (black) compared with Argo floats (a,b,c,d, red) and glider measurements (e,f,g,h, magenta). (i) SST (°C) and (j) SSS (psu) from SYM4 (black) and TSG-Alis data (blue)**

## 4.3 Mixed layer depth

The seasonal distribution of simulated mixed layer depth (MLD) in the SCS basin is evaluated by comparison with values computed from Argo profiles. The MLD is calculated based on a 0.5 °C temperature criteria, corresponding to the temperature difference between the near-surface and the MLD.

Figure 9 shows the winter (DJF) and summer (JJA) spatial distributions of the colocalized simulated (in SYM4) and observed MLD at Argo locations (in space and time), as well as the simulated and observed time series of monthly mean MLD averaged over the Argo points over the SCS. Spatial distributions of SYM4 MLD are close to observed values. Observed and simulated MLD are deeper in winter (varying between ~80 m in the north and ~30 m in the east, Fig. 9a,b) and shallower in summer (varying between ~50 m in the south and ~20 m in the north, Fig. 9d,e). The simulated MLD in both seasons are in general shallower than MLD obtained from Argo profiles, with bias locally reaching 20 m in DJF (Fig. 9c,f). This shallower MLD explains the slight temperature underestimation and salinity overestimation around ~50 m depth (Fig. 9b,d). The average bias over the area and 2010-2018 period is equal to 9.4 m (Fig. 9g), and is stronger for higher values of MLD in winter (e.g. ~-40 m in January 2012). This bias is stable over the 9 years of analysis (Fig. 9g), indicating that there is no drift in terms of simulated MLD. Moreover, the observed temporal variability of MLD is well reproduced by SYM4, with a 0.91 (p<0.01) highly significant correlation between the simulated and observed interannual timeseries of monthly MLD.

Fig. 10f illustrates the climatological seasonal cycle of MLD over the zone 112-118°E, 12-18°N and the period 2010-2016 for all modeled outputs compared to Argo. All models underestimate the MLD (Table 3), but simulate similar annual evolution of MLD, with highly significant correlation between the simulated and observed climatological monthly timeseries (R>0.92, p<0.01). The deepest MLD is observed in winter and the shallowest in April-May. OFES shows the smallest underestimation and NRMSE (7.3 m, 19%), followed by SYM4 (9.10 m, 25%). The strongest biases and NRMSE are obtained from SYM12 (12.5 m, 42%), INDESO (15.6 m, 41%) and COPERNICUS (15.5 m, 40%).

This systematic underestimation of simulated MLD, stronger for higher values of MLD, could be partly related to the underestimation of wind speed. All models use bulk formulae from Large and Yeager (2004), and five of the six simulations compared here use outputs from ECMWF analysis (SYM4 and SYM12, INDESO, COPERNICUS) or reanalysis (GLORYS) as atmospheric forcing, while OFES uses JRA55 outputs (Table 1). Compared to QuikSCAT, ECMWF analysis and reanalysis indeed underestimate sea surface wind speed (by ~1 m.s$^{-1}$ on average over the region for the analysis, Fig. A4). More generally, Herrmann et al. (2020, 2022) showed that global and regional atmospheric models underestimate sea surface wind speed over the SEA region. Wang et al. (2020) showed that both ERA-Interim (produced by ECMWF) and JRA55 underestimate wind over China Seas, with a smaller bias in JRA55 (0.22 m.s$^{-1}$ over the period 1988-2015) than in ECMWF (0.62 m.s$^{-1}$). This underestimation of wind speed in forcing datasets, weaker in JFRA55, partly explains first why all models underestimate the MLD, and second why OFES, which uses JRA55, produces the closest MLD to observations (Fig. 10f). Moreover, as shown by Tréguier et al. (2023), MLD

biases as well as their differences among models may also be due to the model formulation, parameterizations and
resolution, whose shortcomings vary between models: horizontal and vertical resolutions, inclusion of tides, vertical
mixing parameterisation, advection schemes, etc. Gaube et al. (2019) for example showed that mesoscale eddies,
whose representation depends on those formulations, modulate the MLD. Indeed, though SYM4, SYM12 and
COPERNICUS (which provides the initial and lateral oceanic boundary conditions to SYM4) use the same atmospheric
conditions, the MLD underestimation is weaker in SYM4 than in SYM12 and COPERNICUS. This suggests first that
the MLD underestimation in SYM4 can also be explained by the MLD underestimation of the initial and entering
profiles provided by COPERNICUS, and second that SYM4, due to different formulations, in particular its high
resolution, is able to partially correct the stratification of these profiles.

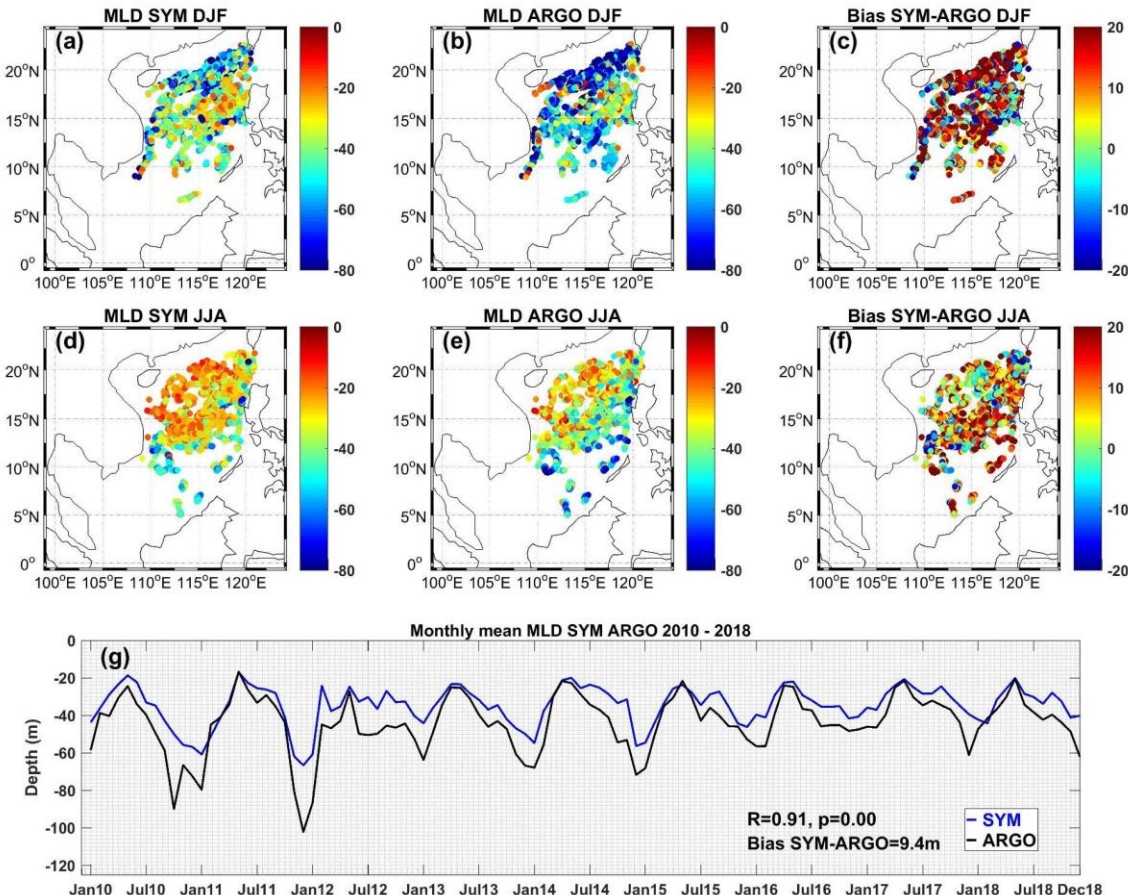

**Figure 9 : Seasonal distribution of MLD (m) from (a,d) SYM4 and (b,e) Argo data and their bias (c,f) in winter (1st row) and summer (2nd row) over 2010-2018; (g) time series of monthly MLD (m) averaged over the domain in SYM4 and Argo data.**

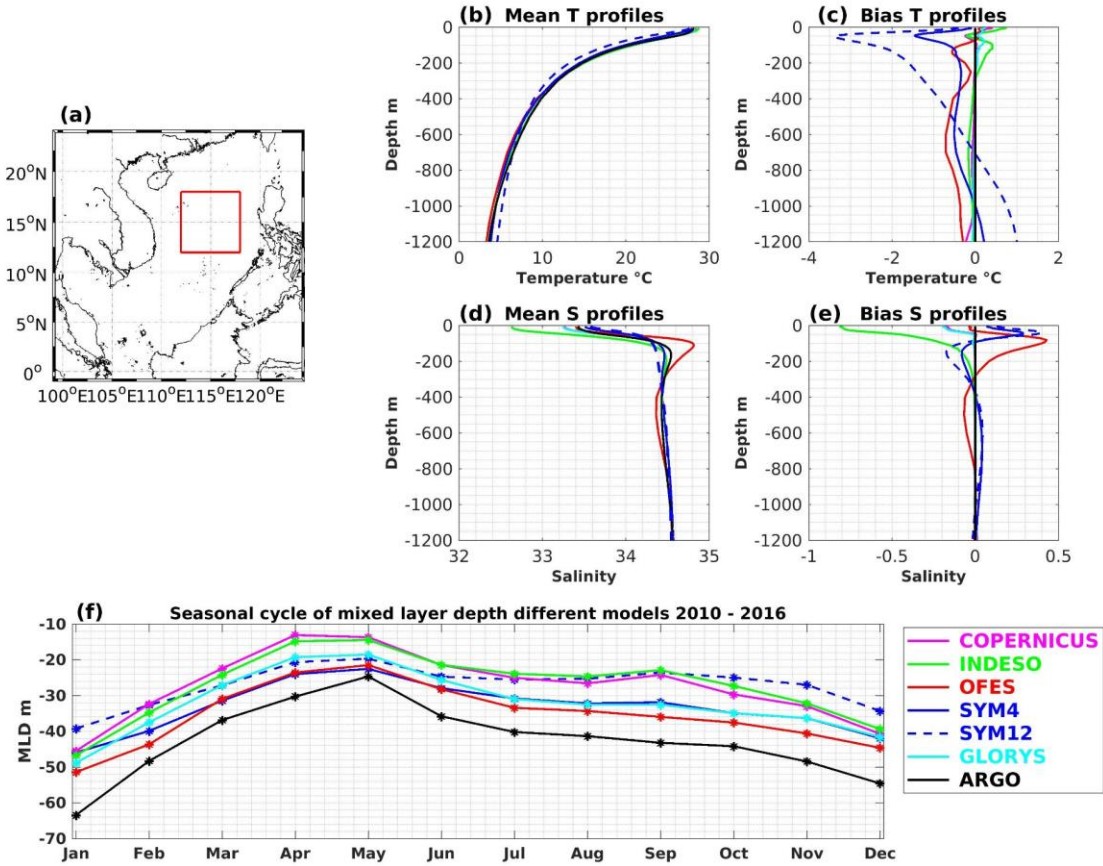

**Figure 10 : (a) Comparison zone ( red box, 112-118°E, 12-18°N); Simulated (SYM4, SYM12, COPERNICUS, INDESO, GLORYS and OFES) and observed (Argo) temperature (b, °C) and salinity (d, psu) profiles averaged over the comparison zone and 2010-2016 and their biases (c,e) compared to Argo; (f) Climatological monthly time series of simulated and observed mixed layer depth over the comparison zone and the period 2010 - 2016.**

## 5. Evaluation and analysis of SCS interocean straits water volume exchanges and SCS water budget

In this part, we first compile the available estimates of fluxes through SCS interocean straits, and use them to assess the ability of our model to reproduce those fluxes. We then use the SYM4 to assess the contributions of lateral interocean, surface atmospheric and river fluxes and of internal variations to the SCS water volume budget at climatological and seasonal scales.

**5.1 Evaluation of simulated water volume fluxes at interocean straits**

A summary of lateral water volume transport published estimates at different SCS interocean straits is presented in Table 4. In-situ measurements are more numerous for Luzon and Taiwan Straits than for Karimata and Mindoro, and no direct measurements are available at Balabac and Malacca. Given the complexity of bathymetry and current conditions in interocean straits, it is difficult to obtain accurate estimates of year-round transport from in-situ observations. In addition to measurements, several models have thus been used to quantify those transports. For a given strait, those studies provided various results, due to differences in methodology (sampling, studied period, choice of transects, model configuration, resolution, parameterisations and forcings…).

**Table 4: Synthesis of lateral water volume transports (in Sv) through SCS straits obtained from previous numerical and observation studies and from our study. Positive = inflow and negative = outflow. When possible, fluxes ratio at different straits are calculated for four main straits (1st column, in order of appearance, Luzon (entrance) : Taiwan:Mindoro:Karimata (exits))**

| References<br>*Fluxes Ratio* | Method / resolution /<br>analyzed period | Luzon | Taiwan | Mindoro | Balabac | Karimata | Malacca |
|---|---|---|---|---|---|---|---|
| Our study<br>***4:1:2:1*** | Model: SYM4 4km ~ 1/28°, 2010-2018 | 4.28±1.59 | -1.21±0.08 | -2.22±1.31 | 0.11±0.23 | -1.00±0.16 | -0.07±0.02 |
| ***5:1:4:1*** | Model: SYM12 12km ~ 0.1°, 2010-2018 | 5.24±1.49 | -1.22±0.09 | -3.58±1.17 | 0.45±0.34 | -0.93±0.15 | -0.08±0.02 |
| Metzger and Hurlburt (1996) | Model: NLOM<br>(Reduced gravity 1.5 layer)<br>1/2° latitude<br>45/64° longitude<br>1982-1983 | 4.4±2.5 | | | | | |
| Qu et al. (2004) | Model: MOM2.0 1/4°<br>1982 - 1998 | 2.4 | | -0.7 | | -1.7 = sum of Karimata and Malacca | |
| Xue et al. (2004) | Model: POM 9-12km<br>20 yr climatology | 2 | | | | | |
| Wu and Hsin (2005) | Model: EAMS 1/8°<br>1999-2003 | | -1.09 | | | | |
| Fang et al. (2005)<br>***10:1:4:3*** | Model: GFDL MOM2.0 1/6°<br>10yr climatology | 4.37 | -0.45 | -1.77 | -0.61 | -1.32 | -0.22 |
| Tozuka et al. (2007)<br>***9:5:1:4*** | Model: MOM3.0 0.4°<br>10yr climatology | 3.6 | -1.8 | -0.4 | | -1.4 | |
| Fang et al. (2009)<br>***4:1.5:1:1*** | Model: MOM2.0 1/6° 1982-2003 | 4.80 | -1.71 | -1.35 | -0.41 | -1.16 | -0.16 |
| Yaremchuk et al. (2009) | Model: Reduced gravity $4^{1/2}$-layer 0.5° | 2.4±0.6 | -0.6±0.5 | -1.5±0.4 | | -0.3±0.5 | |

| References | | Luzon | Taiwan | Mindoro | Balabac | Karimata | Malacca |
|---|---|---|---|---|---|---|---|
| *8:2:5:1* | Upper 750 m | | | | | | |
| Tozuka et al. (2009) | Model: MOM3.0 0.4°- 2° 1980-2006 | 4.4 | | | | -1.6 | |
| Wang et al. (2009) *9:5:3:1* | Model: HYCOM 1/6° 30 yrs climatology | 4.5 | -2.3 | -1.7 | 0.01 | -0.5 (Sunda Shelf transport) | |
| Metzger et al. (2010) | Model: HYCOM 1/12° 2003-2006 | | | | | -0.6 | -0.08 |
| Liu et al. (2010) | Model: MOM4p0d 1/10°-2° 1995-1999 | | -1.88±0.32 | | | | |
| Liu et al. (2011) *3:1:2:1* | Model: BRAN 0.1° - 2° 1993-2008 | 4.81 | -1.44 | -2.27 | 0.01 | -1.42 | -0.27 |
| Xu and Malanotte-Rizzoli (2013) | Model: MITgcm/ -FVCOM 2°/5-18-50km 1960s/1990s | 5.6 | | -2.0 | | -1.4 | |
| Hsin et al. (2012) *5:3:1:1* | Model: EAMS 1/8° 2002-2008 | 4.0±5.1 | -2.6 | -0.9 | 0.1 | -0.8 | |
| Zhang et al. (2014) | Model: 2D barotropic 1/10°-1/30° 2005 - 2008 | | -0.78±1.29 | | | | |
| Tozuka et al. (2015) *30:15:1:23* | Coupled model: UTCM 0.4°-2° OGCM T42 AGCM 160 yrs climatology | 2.9 | -1.5 | -0.1 | 0.8 | -2.3 | |
| He et al. (2015) | Model, BRAN 0.1° 1996-2006 | | | | | -1.6 | |
| Daryaboy et al. (2016) | Model: ROMS 9-50 km 10 yr climatology  Observations:  SODA dataset | | | | | -0.18  -0.29 | -0.14  -0.13 |
| Wei et al. (2016) *7:2:4:1* | Model: POM/ATOP 0.1°x0.1° 2004-2012 | 4.9 | -1.1 | -2.6 | | -0.7 | |
| Wang et al. (2019) *9:3:4:1* | Model: GL-Ba008 dataset (HYCOM) 7km 2004-2014  Observations: ADCP 11/2008-06/2015 | 4.67 | -1.6 | -2.13 | | -0.5  -0.74 (mean | |
| References | Observational studies | Luzon | Taiwan | Mindoro | Balabac | Karimata | Malacca |
| Wyrtki (1961) | Observations: Dynamics method (1909-1957) Upper 175m | 0.5 | | | | | |

| | | | | | | | |
|---|---|---|---|---|---|---|---|
| Qu (2000) | Observations: Dynamics method (WOA1994) Upper 400 m | 3.0 | | | | | |
| Chu and Li (2000) | Observations: GDEM/MOODS dataset 1930-1997 | 6.5 | | | | | |
| Tian et al. (2006) | Observations: LADCP/CTD (Oct 2005) | 6.0±3.0 | | | | | |
| Yuan et al. (2008) | Observations: NCEP/ hydrographic dataset (Aug-Sep 1994) | 3.5 | | | | | |
| Yang et al. (2010) | Observations: LADCP/CTD (Jul 2007) | -5.5 | | | | | |
| Fang et al. (1991) | Observations: Current meters 1980s | | -2 | | | | |
| Chung et al. (2001) | Observations: ADCP/CTD (May, Aug 1999) | | -2.0 (May) -2.2 (Aug) | | | | |
| Wang et al. (2003) | Observations: ADCP (1999-2001) | | -1.8 | | | | |
| Isobe (2008) | Estimation from current observation published previously | | -1.2 | | | | |
| Hu et al. (2010) | Estimation from 30 previous observational and numerical studies | | -2.3 (summer) -0.8 (winter) | | | | |
| Fang et al. (2010) | Observations: ADCP Jan-Feb 2008 | | | | | -3.6±0.8 | |
| Susanto et al. (2013) | Observations: ADCP Dec 2007-Nov 2008 | | | | | -0.5±1.9 | |
| Qu and Song (2009) | Observations: SSH & OBP data (2004-2007) | | | -2.4 | | | |
| Sprintall et al. (2012) | Observations: Mooring ADCP (2008) | | | -0.07 | | | |

Fig. 11 shows the climatological average over 2010-2018 of the lateral fluxes flowing through the SCS interocean straits, the air-sea surface fluxes, the continental river inputs and the internal yearly variations in our simulation.

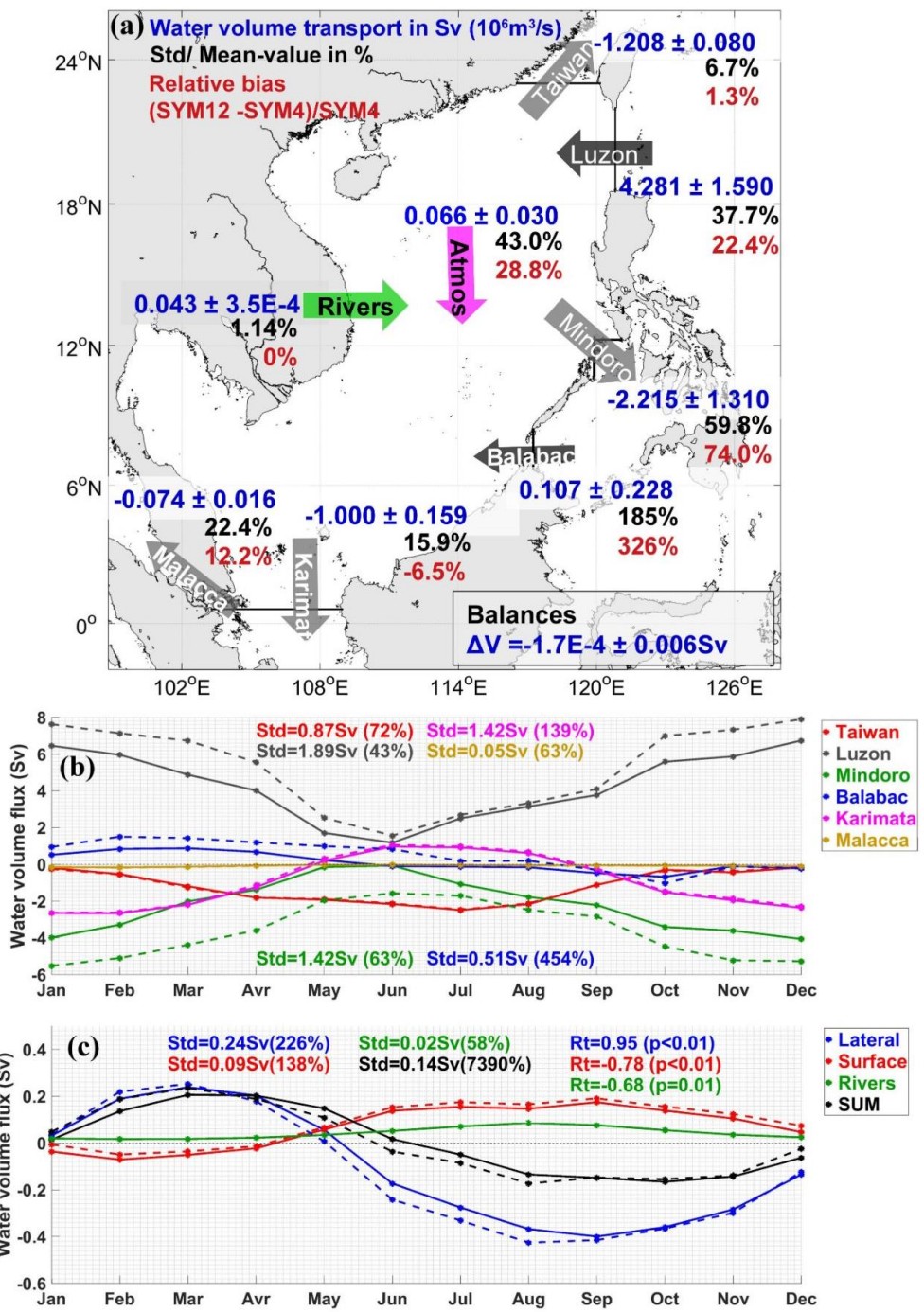

**Figure 11: (a) 2010-2018 averages and standard deviations of water volume yearly fluxes at interocean straits (black arrow = inflow, gray arrow = outflow), from the atmosphere (magenta arrow) and from rivers (green arrow). Positive and negative**

 **values correspond to inflow and outflow, respectively. In black the ratio (in %) of standard deviation and mean value. In red the relative difference (in %) between the absolute value of volume transports in SYM12 and SYM4 : a positive vs. negative value corresponds to a stronger vs. weaker (in or out)-flow in SYM12. The yearly water volume variation in the whole SCS over 2010 - 2018 is provided on the bottom right corner. Time series of monthly averages of (b) lateral water volume transport through each strait and of (c) total lateral, river and atmospheric water volume fluxes and their sum**
 **(equal to the monthly internal variation, Eq. 1) over the SCS domain of SYM4 (solid line) and SYM12 (dashed line). Std and Rt stand for standard deviation and correlation with the sum of fluxes, respectively, calculated for SYM4.**

### 5.1.1 Climatological mean values and seasonal cycle

In SYM4, seawater enters the SCS through the Luzon and Balabac Straits and flows out of the basin via the Taiwan, Mindoro, Karimata and Malacca Straits, forming the SCSTF identified by Qu et al. (2005) (Fig. 11a). The main inflow into the SCS is from the Pacific Ocean through the Luzon Strait, with an average simulated water volume inflow of 4.3 Sv, in the range of LST previous numerical estimates that vary between 2.4 and 4.8 Sv (Metzger and Hurlburt 1996, Qu et al. 2004, Xue et al. 2004, Tozuka et al. 2007, 2009, 2015, Fang et al. 2005, 2009, Liu et al. 2011, Hsin et al. 2012, Xu and Malanotte-Rizzoli 2013, Wei et al. 2016, Wang et al. 2019 - Table 4) and from previous observational studies, that vary between 3.0 and 6.5 Sv (Qu 2000, Chu and Li 2000 - Table 4).

Fig. 11b illustrates the simulated seasonal cycle of water volume transport through SCS interocean straits in SYM4. The LST (black line) is positive (westward) all over the year, with a maximum intrusion in winter (December, 6.74 Sv) and a minimum intrusion in summer (June, 1.18 Sv), in agreement with previous numerical studies listed in Table 4 (Metzger and Hurlburt 1996, Qu 2000, Qu et al. 2004, Chu and Li 2000, Fang et al. 2005, Yaremchuk et al. 2009, Liu et al. 2011, Xu and Malanotte-Rizzoli 2013, Wang et al. 2019). Our model averaged LST in January (6.46 Sv), August (3.14 Sv) and October (5.58 Sv) is also close to observations of respectively Qu (2000, 5.3 Sv in January - climatology value), Yuan et al. (2008, 3.5 Sv in August 1994), and Tian et al. (2006, 6±3 Sv in October 2005).

SYM4 simulates a 1.21 Sv water volume outflow through the Taiwan Strait. This is consistent with numerical outflow estimates varying from 0.45 to 2.6 Sv (Wu and Hsin 2005, Fang et al. 2005, 2009, Tozuka et al. 2007, 2015, Yaremchuk et al. 2009, Wang et al. 2009, Hu et al. 2010, Liu et al. 2011, Hsin et al. 2012, Zhang et al. 2014, Wei et al. 2016, Wang et al. 2019, Table 4), as well as with observational estimates of 1.8 - 2.0 Sv (Fang et al. 1991, Wang et al. 2003, Isobe 2008 - Table 4). This transport is negative the whole year (Fig. 11b, red line), with a maximum 2.50 Sv outflow in July, a decrease in autumn - winter until a minimum 0.16 Sv outflow in December. These results are consistent with numerical estimations of Xue et al. (2004), Fang et al. (2005, 2009), Yaremchuk et al. (2009), Liu et al. (2011), Zhang et al. (2014), Wang et al. (2019) and close to measurements of Wang et al. (2003) who found a maximum outflow of 2.7 Sv in summer and a minimum outflow of 0.9 Sv in winter.

At Mindoro Strait, SYM4 simulates an average lateral seawater outflow of 2.22 Sv, in agreement with estimates from previous modeling studies varying from 0.1 to 2.6 Sv (Liu et al. 2011, Wang et al. 2019, Fang et al. 2009, Yaremchuk et al. 2009, Xu and Malanotte-Rizzoli 2013, Tozuka et al. 2007, 2015 - Table 4). The simulated outflow is also quite

close to observations (2.4 Sv) analyzed by Qu and Song (2009), and stronger than in-situ estimates (0.07 Sv) of
Sprintall et al. (2012). Similarly to previous studies mentioned above, SYM4 simulates an outflow at Mindoro for the
whole year (Fig. 11b, green line), maximum in December (4.05 Sv) and minimum in summer (0.06 Sv outflow in
June).

In the south of the SCS, SYM4 produces a 1.0 Sv seawater outflow through the Karimata Strait, in agreement with
previous numerical estimates ranging from 0.3 to 2.3 Sv (Fang et al. 2005, 2009, Yaremchuk et al. 2009, Tozuka et
al. 2007, 2009, 2015, Wang et al. 2009, Metzger et al. 2010, Liu et al. 2011, Xu and Malanotte-Rizzoli 2013, He et al.
2015, Daryaboy et al. 2016, Wei et al. 2016, Wang et al. 2019, Table 4) and slightly larger than estimates from
measurements (0.50 to 0.74 Sv). The simulated annual cycle of the Karimata water volume transport (Fig. 11b,
magenta line) shows a southward outflow from September to April, and a northward inflow from May to August, with
values ranging from -2.64 Sv in January to 0.97 Sv in June, again in agreement with previous studies (Fang et al. 2005,
2009, Xue et al. 2004, Yaremchuk et al. 2009, Liu et al. 2011, Xu and Malanotte-Rizzoli 2013, He et al. 2015, Wei et
al. 2016, Wang et al. 2019, Daryaboy et al. 2016, Susanto et al. 2013, Wang et al. 2019).

Compared to the four main straits of the SCS, Balabac and Malacca Straits show water volume transports one order
of magnitude smaller (Fig. 11b), and observational studies are scarce. SYM4 produces an annual mean westward
inflow of 0.11 Sv at Balabac, in agreement with the estimate of 0.1 Sv by Hsin et al. (2012), while Wang et al. (2009)
and Liu et al. (2011) estimated very small inflows (0.01 Sv) and Tozuka et al. (2015) suggested a much stronger inflow
(0.8 Sv). Fang et al. (2005, 2009), in contrast, proposed an eastward outflow of, respectively, 0.061 Sv and 0.41 Sv at
these channels. In SYM4, water enters at Balabac from January to May (maximum of 0.88 Sv in March, Fig. 11b,
blue line), and exits from June to December (maximum of 0.68 Sv in October). Fang et al. (2005, 2009), on the other
hand, obtained an outflow the whole year.

At the narrowest interocean strait of the SCS, Malacca, water flows out of the SCS toward the Andaman Sea (Indian
Ocean) at a rate of 0.07 Sv in our simulation, at the low end of the range of previous estimates (from 0.08 to 0.27 Sv
- Metzger et al. 2010, Fang et al. 2009, Liu et al. 2011, Daryaboy et al. 2016). The Andaman Sea receives water from
the SCS all over the year in SYM4 (maximum 0.16 Sv in February, minimum 0.02 Sv in May and June, Fig. 11b,
cyan line) consistently with model results of Fang et al. (2005, 2009).

Last, the water volume transport at Balabac Strait shows the strongest annual variability over the yearly cycle (Fig.
11b), with a very high standard deviation relative to the average (0.51 Sv, 454%), followed by Karimata (1.42 Sv,
139%), Taiwan (0.87 Sv, 72%), Mindoro (1.42 Sv, 63 %), Malacca (0.05 Sv, 63%), and Luzon (1.89 Sv, 43%) Straits.

**5.**1**.2 Vertical structure**

We examine the vertical distribution of water volume interocean fluxes along the water column for the whole year,
summer and winter in SYM4 (Fig. 12).

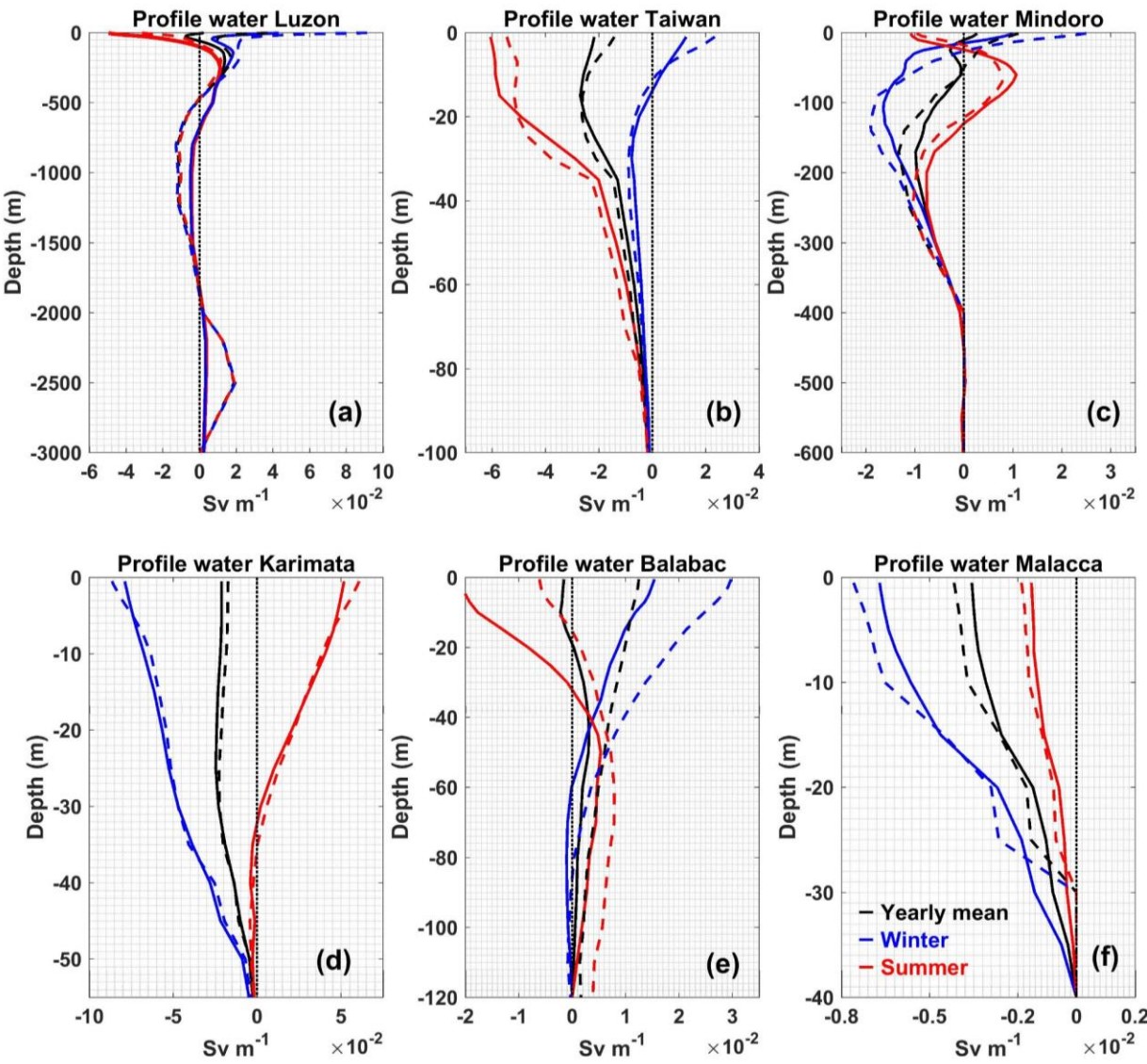

**Figure 12: Yearly (in black) and seasonal (winter: December, January and February, in blue and summer: June, July, August, in red) climatological averages over 2010-2018 of vertical profiles of water volume lateral fluxes via the 6 interocean straits (Sv.m⁻¹) in SYM4 (solid line) and SYM12 (dashed line). Positive, resp. negative values indicate inflow, resp. outflow. The depth axis varies and is adapted to the depth of the strait.**

At the largest and deepest strait of the SCS, the Luzon Strait, SYM4 simulates a strong seasonal variability of fluxes in the surface and subsurface layers, from 0 to 400 m (Fig. 12a). In the first 50 m, the lateral seawater flux is westward (inflow) in winter and eastward (outflow) in summer, in phase with the seasonal wind forcing (northeast monsoon winds in winter and southwest monsoon winds in summer). This seasonal variability of the surface flow is consistent with observations of Centurioni et al. (2004) obtained from Argo floats data showing an inflow in the upper 15 m from the Pacific to the SCS in winter, but no inflow in summer. The yearly averaged flux is eastward for the upper 50

m. In the 50-700 m layer, summer, winter, and yearly mean fluxes are all westward, with a maximum inflow between 100 and 300 m. Until 400 m depth, inflow is stronger in winter, and weaker in summer. Below 400 m depth, we do not obtain any significant seasonal variability. SYM4 produces an outflow for all seasons between 700 and 1900 m (slightly stronger in winter), then a weak inflow in the deep layer (slightly stronger in summer), from 1900 m until the bottom. Qu et al. (2004), Hsin et al. (2012), Nan et al. (2013, 2015), Liu and Gan (2017) found the same vertical structure of water volume transport crossing the Luzon Strait using numerical methods, as well as Li and Qu (2006) using dissolved oxygen distribution data.

The yearly mean water volume fluxes through the Taiwan Strait are negative, i.e. northward, along the whole water column (Fig. 12b). SYM4 simulates a strong seasonal variability of vertical distribution of fluxes at this shallow strait, again triggered by the atmospheric forcing. In winter, under the northeast monsoon wind, fluxes flow southward in the surface layer (0-10 m), then underneath this depth, fluxes become northwards. In summer, under the southwest monsoon wind, fluxes are northwards from the surface to the bottom, and particularly strong between 0 and 15 m.

Regarding the Mindoro Strait (Fig. 12c), SYM4 simulates a strong seasonal variability of fluxes for the upper 300 m. Deeper than 300 m, we observe no significant seasonal variability of fluxes, which are negligible below 400 m. In winter, fluxes flow inward in the layer 0-20 m, and outward between 20 and 400 m, with a maximum outflow around 120 m. In summer, we obtain a "sandwich" vertical distribution like at the Luzon Strait, with outflows in the upper layer (0-30 m, maximum at the surface) and in the subsurface layer (130-400 m, maximum near 200-250 m) and an inflow between these two layers (30-130 m, maximum near 60 m). Again, winter westward inflow and summer eastward outflow in the surface layer follow the monsoon wind direction. Below 130 m, both winter and summer fluxes flow outward, but the winter outflow is much stronger than the summer one. The annual mean flux shows a small inflow in the first 10 m and an outflow below 10 m, with local maxima at 30 and 180 m depths.

At the shallow Karimata Strait (Fig. 12d), the yearly climatological fluxes are southwards (outflow) over the full depth. Like for the Taiwan Strait, fluxes through the Karimata Strait strongly vary with the seasonal monsoon summer southwest and winter northeast winds. In summer, fluxes enter the basin above 30 m depth, then slightly flow out in the deepest layer until 55 m depth. In winter, fluxes are southwards all along the water column. This simulated seasonal variability of vertical fluxes is in agreement with *in-situ* measurements of Fang et al. (2010), who found that the monsoon was the main factor influencing the fluxes at the Karimata Strait.

The vertical structure of fluxes across the shallow Balabac Strait also connecting the SCS with the Sulu Sea also varies strongly with the seasonal cycle (Fig. 12e). In winter, the (westward) inflow is maximum at the surface then decreases with depth. From 60 m depth to the bottom, the winter flux becomes slightly negative. The situation is opposite in summer: fluxes flow eastward (outflow) in the surface layer with a decrease with depth until 30 m, then flow westward from 30 m depth until the bottom. The annual mean fluxes are negative for the upper 20 m, then positive until the bottom (with a maximum inflow at 40 m depth).

Fluxes crossing the Malacca Strait flow westward (outflow) all year round along the whole water column, with stronger values near the surface, and a decrease with depth, and stronger fluxes in winter (Fig. 12f). As the section is very shallow (40 m depth), the monsoon wind again plays an important role in the difference of seasonal fluxes intensity: the northeast wind reinforces the outflow in winter, and the southwest monsoon reduces it in summer.

**5.2 Contributions of surface, river and lateral interocean fluxes to the SCS water volume budget**

In this section we analyze the water volume budget obtained from our SYM4 simulation and quantify the contributions of each term of Eq. 1 to the internal variations of water volume in the SCS: lateral fluxes flowing through the SCS interocean straits, air-sea surface fluxes and continental inputs from rivers.

The average net seawater volume exchanged through SCS interocean straits over the period 2010 - 2018 in SYM4 is equal to -0.11 Sv (Fig. 11a). This net lateral loss at the domain oceanic open boundaries is balanced by the inputs from
rivers and atmosphere, evaluated respectively at 0.04 Sv and 0.07 Sv, i.e. a total freshwater volume input of 0.11 Sv (Fig. 11a). The difference between the gain from atmosphere and rivers and the loss from straits, i.e. the water volume variation, is equal to -1.7e-4 Sv, equivalent to a decrease of sea level of 1.6 mm year$^{-1}$ over the period 2010-2018 and negligible compared to the total water volume input (0.004%). SYM4 value of atmospheric freshwater flux is slightly smaller than estimates of Qu et al. (2006) who provided an annual mean value of 0.1 - 0.2 Sv through analyzes of
several dataset of precipitation (CMAP, GPCP, TRMM) and evaporation (NCEP reanalysis). Fang et al. (2009) deduced from the land discharges relation of Perry et al. (1996) an annual river flux of 0.05 Sv, close to our yearly river water volume flux. Using numerical models, Qu et al. (2006), Fang et al. (2009) and Wang et al. (2019) obtained a yearly average of total freshwater flux over the whole SCS respectively of 0.08 Sv (period 1950 - 2003), 0.11 Sv (period 1982 - 2003) and 0.11 Sv (period 2008 - 2015), quite close to SYM4 result. These previous studies assumed
a null total lateral water volume flux and calculated the total freshwater flux based on the lateral salt budget and the mean salinity of the whole basin.

Fig. 11c illustrates the 10-year climatological monthly averages of all the water volume fluxes exchanged between the SCS and surrounding environment in SYM4: the total interocean lateral flux, river runoff, surface atmospheric flux, and the sum of all three fluxes, which equals the internal water volume variation (Eq. 1). The total lateral flux is
positive (inflows > outflows) from January to May and negative (inflows < outflows) the rest of the year. The strongest net lateral inflow occurs in March (0.24 Sv) and the strongest outflow in September (-0.40 Sv). Atmospheric and river fluxes are out of phase with those lateral fluxes. Surface fluxes are negative during the winter - spring dry season (with a maximum surface loss in February, -0.07 Sv) when evaporation dominates precipitation. During the summer - autumn rainy season, precipitation becomes abundant, the surface fluxes become positive (maximum value in
September 0.17 Sv) and the freshwater river flux increases from May to October with a maximum in August (0.09 Sv).

To better understand the role of the SCS in the global and regional water cycle, we analyze the sum of the water volume fluxes (Fig. 11c, black curve) exchanged yearly over the domain, equal to the water volume variation. In overall, the SCS stores water from January to June (with a maximum 0.21 Sv storage in March and April) and releases

water from July to December (minimum of -0.17 Sv in October). The correlations between monthly oceanic lateral, surface and river fluxes and the total water monthly flux are respectively 0.95 (p<0.01), -0.78 (p<0.01) and -0.68 (p=0.02). The lateral winter gain (and summer lateral loss) largely exceeds the atmospheric winter loss (and summer atmospheric and river gain). Moreover, the standard deviation of the climatological monthly total lateral flux (0.24 Sv) is about three times higher than the standard deviation of atmospheric fluxes (0.09 Sv) and 10 times higher than

the standard deviation of river fluxes (0.02 Sv). Over an annual cycle, the monthly variability of the lateral flux, which is strongly driven by monsoon winds, therefore dominates the variability of the two other fluxes (atmospheric and river) and drives the annual cycle of the SCS water storage. The low variability of river fluxes compared to lateral and surface fluxes is partly explained by the use of monthly climatology river runoff for most of the rivers in SYM4, especially for huge rivers such as the Mekong and Pearl rivers.

**5.3. Influence of model resolution**

We showed in Section 4 that the use of a higher resolution model improves the quality of the simulation in terms of ocean dynamics and water masses representation. Here we quantify the influence of this resolution on the water budget estimate. The interocean water volume transports and total water budgets computed from SYM12 are shown in Fig. 11 and Fig. 12.

The direction of averaged yearly water volume transports (inflow or outflow) are the same at all straits for SYM4 and SYM12 simulations (Fig. 11a). However, except for Karimata strait (with a 6.5% decrease of outflow), water volume exchanges are stronger in SYM12 than in SYM4 (from +1% to +70%), in particular at the two main straits: +22% for the Luzon inflow and +70% for the Mindoro outflow. Luzon inflow and Mindoro outflow are especially stronger in March (with respective differences compared to SYM4 of 1.8 Sv and 2.3 Sv, Fig. 11b), and over the whole water

column (Fig. 12a, c, e). Conversely, differences are negligible at Taiwan, Karimata and Malacca straits (< 10%, Fig. 11a), throughout the year (Fig. 11b) and the column (Fig. 12b, d, f). Taiwan, Karimata and Malacca are shallow straits with depth < 100 m, while Luzon and Mindoro are much deeper straits. Though bathymetries of shallow and wide straits are not strongly affected by the resolution of the grid, the differences are considerable in deeper straits, especially in the deep narrow layers, which can partly explain this fluxes overestimation (Fig. A1, A2).

In terms of water budget, we obtain similar seasonal cycles of lateral, surface and river water volume fluxes in SYM12 and SYM4 but slightly though significantly different values (Fig. 11c). SYM12 surface water exchanges are slightly smaller from January to April (when the ocean loses water to the atmosphere), stronger the rest of the year (when the sea receives water), and the net atmospheric input is larger on a yearly average (29%). This difference can be explained by the use of Large and Yearger (2004) bulk formulae: the colder SST (but same air temperature from ECMWF) in

SYM12 compared to SYM4 (Fig. 7a2,a8) results in weaker latent heat flux, hence evaporation. The total interocean lateral water outflow is consequently also larger in SYM12 all year long (+18%), compensating for the larger atmospheric net input.

## 6. Conclusion

The three-dimensional hydrodynamic model SYMPHONIE was implemented over the South China Sea with high
horizontal resolution and an explicit representation of tides to simulate and study the functioning, variability and influence of ocean circulation in the SCS and their role in regional climate.

A simulation was performed at 4 km resolution over the recent 9-year period 2010 – 2018, using three hourly atmospheric forcing, daily lateral oceanic boundary forcing, nine tidal forcing components and real-time or climatology data for 63 river discharge points. The ability of the model to reproduce the characteristics of circulation
and water masses over the SCS, including tides, was evaluated through a thorough comparison with available satellite and in-situ observation datasets and with simulations performed with other models at coarser resolution. The model shows good performances in terms of tidal cycle and of seasonal cycle, interannual variability and spatial distribution of surface characteristics and circulation (SST, SSS, SLA and associated geostrophic currents). Comparisons with Argo and glider profiles and other models showed that SYM4 reproduces well the vertical distributions of temperature
and salinity as well as MLD. These comparisons with observational data and other models and simulations at coarser resolution therefore quantitatively show the added-value of this high-resolution hydrodynamic model that includes tides for the representation and study of the spatial and temporal variability of the SCS dynamics and water masses.

One of the first objectives of this numerical tool is to study the variability of the water volume, heat and salt budgets at different scales, precisely quantifying the contribution of each component: lateral oceanic, surface atmospheric and
river fluxes, internal variations. We implemented an online computation method allowing to rigorously close those budgets: over any given period, and for all the quantities studied, the sum of all fluxes is equal to the variation of the quantity over the period. We quantitatively demonstrated the added-value of the online method. With offline computation based on daily outputs, NRMSE reaches 10 to 30% for interannual variations of yearly values of heat and salt net lateral fluxes. Moreover, the online method allows to rigorously compute at each lateral strait the total
inflowing and outflowing fluxes, contrary to the offline method that induces errors of the same order or even one order of magnitude larger than the values themselves.

This simulation was then used to provide a new quantitative insight on the SCSTF and SCS water volume budgets at the seasonal and climatological scales. Estimates over the 2010-2018 period of the water volume transports through SCS interocean straits and of their seasonal cycle and vertical distributions, and of each term of the volume budget
over the whole domain were examined and compared with a synthesis of previous estimates. They were shown to be in agreement with available observational and numerical studies. According to our simulation, the SCS receives over the period 2010 – 2018 an annual average 4.50 Sv water input, mainly from the Western Pacific through the Luzon

Strait (4.28 Sv, 95%). The remaining input comes from the Sulu Sea through the Balabac Strait (3%), from the atmosphere (1%) and from rivers (1%). The SCS releases all of this water through the other straits: half of the water flows to the Sulu Sea through the Mindoro Strait, about one quarter flows northward to the East China Sea via the Taiwan Strait (27%), and one quarter flows southwestward to the Java Sea through the Karimata Strait (22%). Those results are in agreement with previous modeling studies, who all reported Luzon as the main entrance and Taiwan, Mindoro and Karimatas as the main exits (see ratios provided in Table 3), though differences between quantitative estimates due to different methodologies exist. Our ratios are close to those of Fang et al. (2005), Liu et al. (2011), Wei et al. (2016) and Wang et al. (2019) who all reported Mindoro as the main exit: they estimated the distribution ranges for outflows through the Taiwan, Mindoro and Karimata straits at 10%-34%, 44%-53% and 11%-30%, respectively. Tozuka et al. (2007), Fang et al. (2009), Wang et al. (2009), Hsin et al. (2012) reported the main exit at Taiwan strait (with respective estimates of 35%-60%, 11%-37% and 11-39%), and Tozuka et al. (2015) at Karimata strait (60%). The numerical approach valuably complements in-situ observations, whose spatial and temporal coverage does not currently allow these ratios to be estimated. Obtaining in-situ based estimates is indeed challenging, requiring to implement simultaneous measurements in all these straits over a long period, but would enable further assessment of the robustness of numerical estimates.

Taking the sum of fluxes through the Mindoro, Balabac, Karimata and Malacca Straits (following Fang et al. 2009, He et al. 2015, Wei et al. 2016), we provide a 3.2 Sv estimate for the water volume transport from the SCS into the Indian Ocean: 70% of the total water input to the SCS is transferred toward the Indian Ocean. The analysis of the seasonal cycle of SCSTF and SCS water budgets shows that from February to July, the SCS stores water. This water gain first comes from lateral fluxes (February to April), mainly through the Luzon Strait, then to a lesser extent from rivers and atmospheric freshwater fluxes (May to July). From August to January, the SCS loses water: it receives water from the atmosphere and rivers but releases a larger amount of water through lateral fluxes, mostly through the Taiwan, Mindoro and Karimata Straits, with a peak outflow in August-September. The SCS is a source of water to the atmosphere from January to April (evaporation exceeds precipitation), and a sink from May to December. Interocean exchanges, as well as their vertical structures, show seasonal variations that weaken when depth increases and are driven by monsoon winds. Exchanges at Luzon, Mindoro and Karimata Straits are enhanced during the autumn-winter period, from October to February, and weakened during the spring summer period, from April to September (and even reversed for Karimata), due to the opposite effects of the winter northeast monsoon winds (that favor eastward and southward flows) and of the southwest monsoon winds (that favor westward and southward flows). The situation is the opposite at Taiwan Strait, where the winter northeast monsoon weakens the northward outflow whereas the summer southwest monsoon enhances it. Exchanges through Luzon Strait deep layers show a stable sandwiched structure with vertically alternating inflows and outflows. Finally, seasonal variations of SCS water content are completely driven by the lateral oceanic water volume fluxes through interocean straits, themselves driven by seasonal monsoons.

We showed that the use of a high vs. coarse resolution model improves the quality of the simulation of SCS ocean dynamics and influences the water budget estimate. This 9-year SCS simulation from a high-resolution model producing a consistent closed water volume, heat and salt budgets will now be used to examine in detail the interannual variability of those budgets over the region. Except at Taiwan Strait, interocean and atmospheric water volume fluxes indeed vary strongly on an interannual time scale: standard deviations of yearly fluxes can be of the same order of magnitude as mean values, reaching 38% and 60% at Luzon and Mindoro Straits, and even 185% at Balabac Strait (Fig. 11a). These important variations partly explain the uncertainties of numerical and observational estimates. In addition, a decrease of SSS over the period 2011-2012 followed by an increase until 2016 were observed over the SCS by Zeng et al. (2014, 2018), and attributed to an increase of precipitations and a reduced intrusion of the Kuroshio salty water mass at Luzon Strait, followed by a decrease of precipitations and a stronger Luzon inflow. Those SSS interannual variations are well reproduced by our simulation, that will be used to examine and explain them in detail.

The simulation presented here is fully available to the interested scientific community. Using this hydrodynamical numerical tool to model and understand the SCS ocean dynamics will allow us to examine their influence on other compartments of the regional system. A coupling with a biogeochemical model (Herrmann et al. 2014, 2017) would allow to study the functioning and variability of SCS planktonic ecosystem, which are strongly influenced by ocean dynamics (Bombar et al. 2010; Loisel et al. 2017; Lu et al. 2018). This simulation, or simulation performed over given periods of interests, could also be used to assess the dispersion of potential contaminants over the area (plastics, radioactive contaminants, etc. e.g., Estournel et al. 2012). A coupling with a regional atmospheric model, that will allow to consider and examine the contributions of air-sea interactions in the ocean and atmosphere dynamics in the region, is also under development over the Southeast Asia region to be integrated in the framework of the CORDEX-SEA project (Tangang et al. 2020; Herrmann et al. 2022).

## 7. Appendices

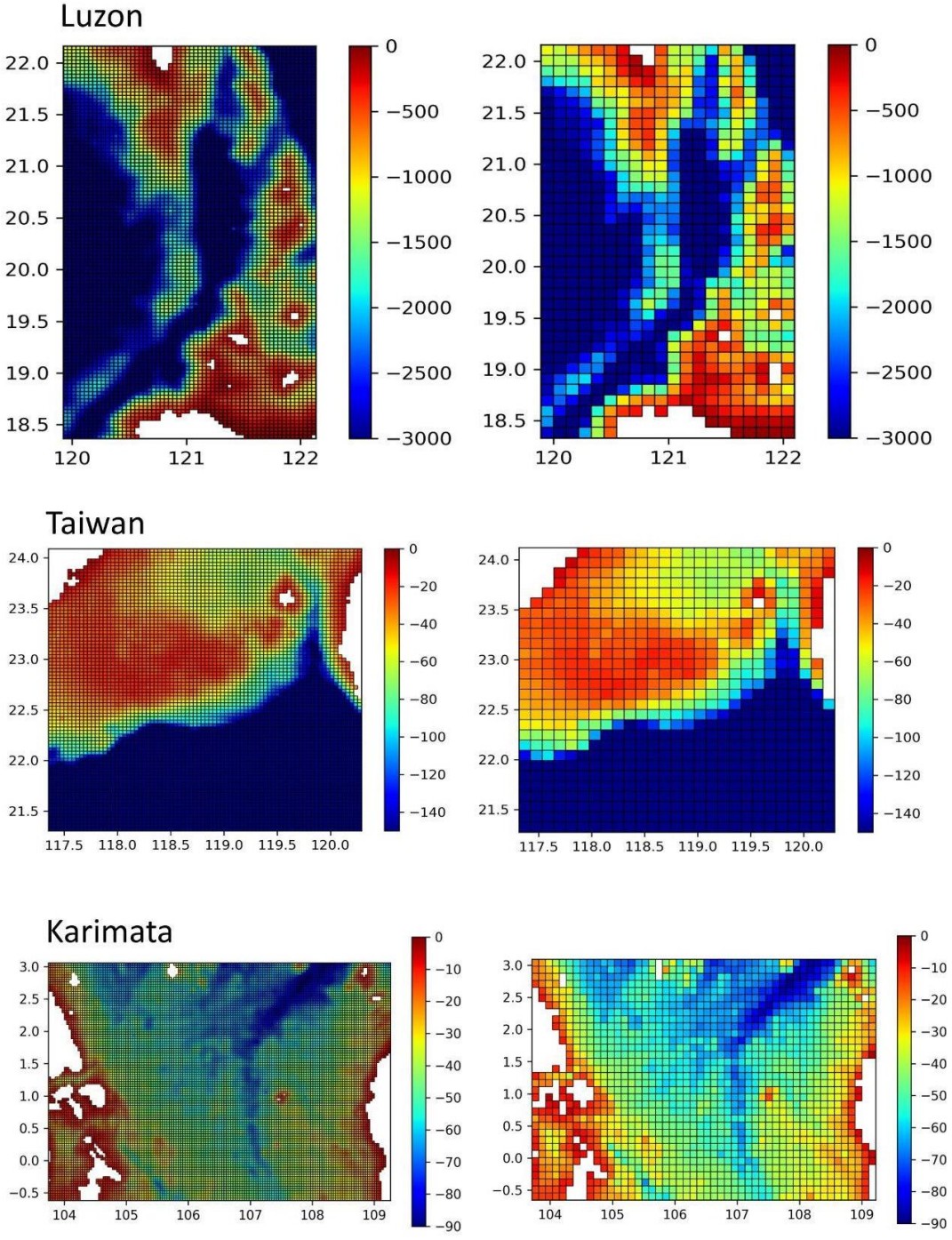

**Figure A1: Bathymetries (m) and grids of interocean straits in SYM4 (left) and SYM12 (right) configurations for Luzon, Taiwan and Karimata straits.**

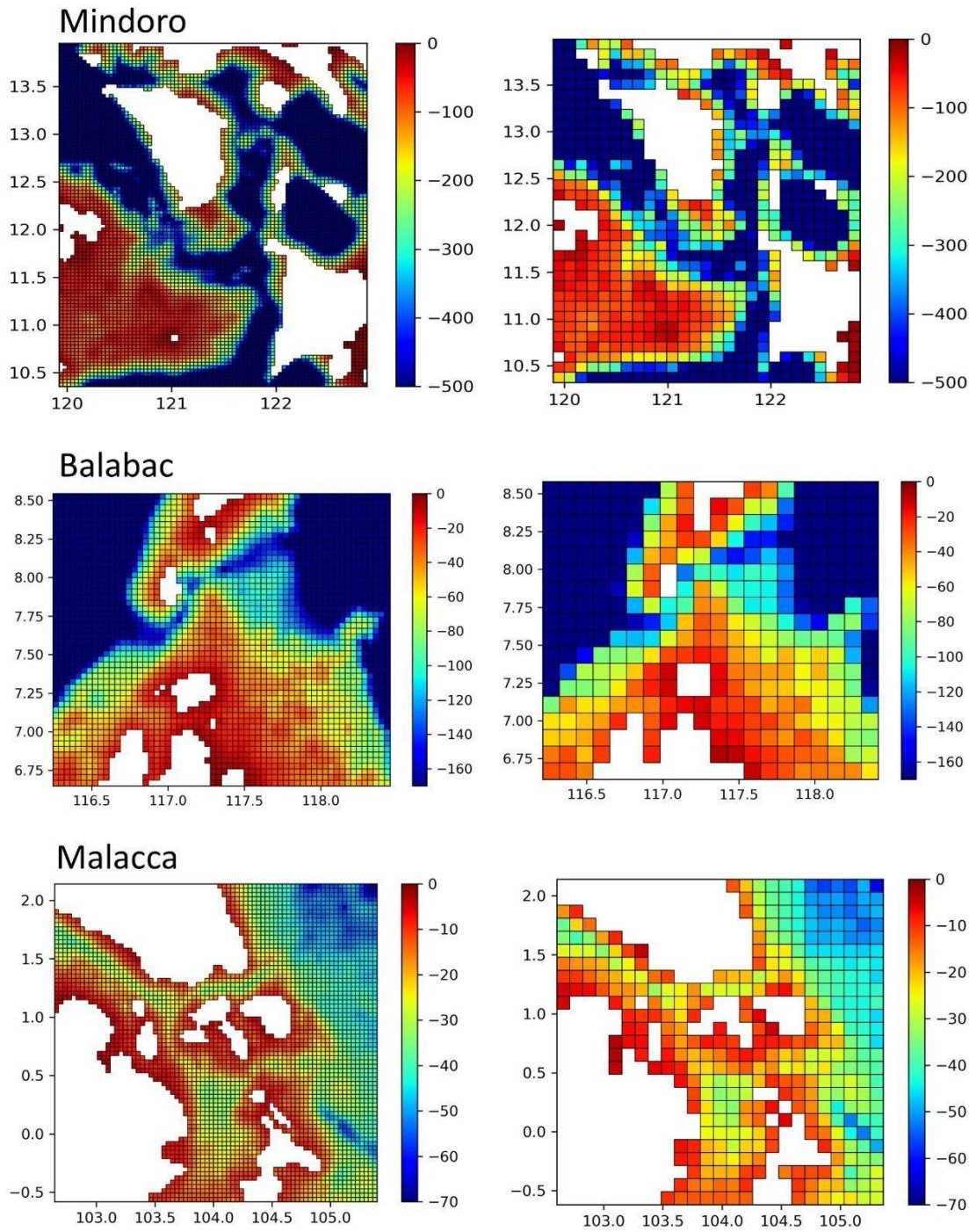

**Figure A2: Same as A1 for Mindoro, Balabac and Malacca straits.**

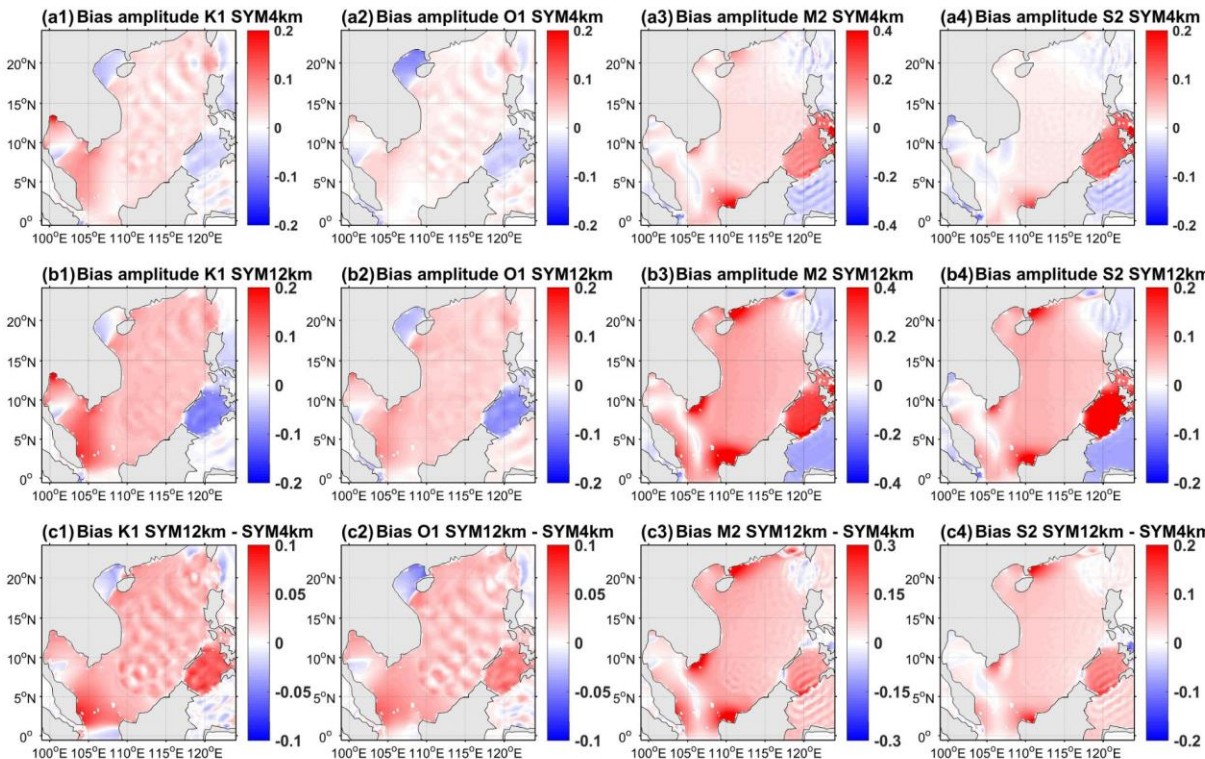

Figure A3: Bias of tidal amplitude (a1-a4) between SYM4 and FES2014, (b1-b4) between SYM4 and FES2014, and (c1-c4) difference of absolute values of SYM12 and SYM4 biases for the 4 tidal components K1, O1, M2, S2.

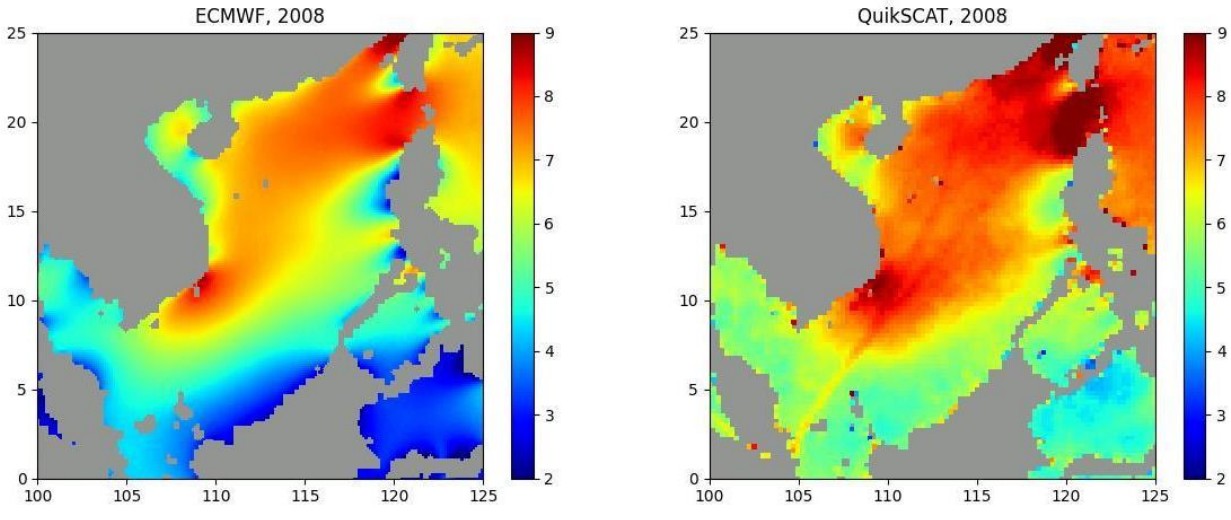

Figure A4: Sea surface wind speed (m s-1) over the SCS averaged over the year 2008 computed from ECMWF 3-hourly analysis outputs (left) and daily QuikSCAT outputs (right).

## Code and data availability

The SYMPHONIE hydrodynamical ocean model version 2.4, the SCS configuration, input files, data for model assessment and code used to generate figures are freely available on https://doi.org/10.5281/zenodo.7941495 (Trinh et al. 2023).

## Author contribution

Trinh Bich Ngoc, Marine Herrmann and Caroline Ulses designed the experiments and Trinh Bich Ngoc carried them out, with the support of Thomas Duhaut, Patrick Marsaleix and Claude Estournel. Patrick Marsaleix, Thomas Duhaut and Claude Estournel developed the model code. Trinh Bich Ngoc, To Duy Thai, and Patrick Marsaleix worked on the model calibration and optimization. Trinh Bich Ngoc, Claude Estournel and Patrick Marsaleix implemented and analyzed the online and offline computational methods. R. Kipp Shearman organized the seaglider survey. Trinh Bich Ngoc, Marine Herrmann and Caroline Ulses prepared the manuscript with contributions from all coauthors.

## Competing interest

The authors declare that they have no conflict of interest.

## Acknowledgments

This work is a part of LOTUS international joint laboratory (lotus.usth.edu.vn) funded by IRD. Numerical simulations were performed using CALMIP HPC facilities (projects P13120 and p20055) and the cluster OCCIGEN from the CINES group (project DARI A0080110098). This work is also supported by the Vietnam Academy of Science and Technology, grant code CSCL17.03/23-24. The authors would like to thank the SIROCCO service (https://sirocco.obs-mip.fr/), coordinated by the Research Infrastructure ILICO (CNRS-IFREMER), for providing the SYMPHONIE code.

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
