# Peer review of "New insights on the South China Sea Throughflow and its seasonal cycle: evaluation and analysis of a configuration at highresolution including tides of the online closed-budget regional ocean model SYMPHONIE version 2.4"

_EGUsphere, 2023_

## Author Comment (AC1)

**Answer to reviewer 1**

The authors utilized the SYMPHONIE model's kernel to conduct a simulation of higher resolution (4 km) compared to previous studies. They evaluated the performance of this simulation by comparing it with satellite and field measurements. The errors in online and offline computation of lateral fluxes were estimated. The logical flow of the manuscript is clear. However, based on my evaluation, this simulation did not provide new and insightful information about the dynamics of this large-area marginal sea. Therefore, I cannot support accepting this research in its current stage. I would like to highlight the following major concerns for the authors' consideration:

We warmly thank the reviewer for the time and attention devoted to our paper, and for those positive and constructive comments. We have carefully considered all the comments and suggestions in the revised version of our manuscript. In what follows, and in the highlighted version of the manuscript, our answers and modifications are highlighted in blue. Line numbers refer to the highlighted version of the revised manuscript.

1) The authors claim that this simulation benefits from higher horizontal resolution, but it is unclear how. Were frequently used models like HYCOM, GLORYS12V1 from CMEMS, and OFES simulation shown to perform poorly compared to the regional simulation presented in the manuscript? It is essential to thoroughly compare this simulation with frequently used models, particularly when the circulations in the study area, such as the South China Sea (SCS), are influenced by complex internal and external forces. If these later simulations performed better than the configured one, I don't think a publication of this manuscript will contribute to the community.

Several groups indeed develop and distribute global or regional simulations that cover the SCS region from other models. Some of those simulations (for example reanalysis and analysis produced by CMEMS and most of HYCOM simulations used to study the area, e.g. Yang et al. 2019, Zithao et al. 2021) include assimilation procedures toward satellite sea surface temperature and elevation data and ARGO temperature and salinity profiles. This helps them to realistically reproduce ocean surface characteristics and water masses profiles as well as their variability, but does not let them completely free to produce their own physics. Conversely, simulations without assimilation (e.g OFES simulations produced by JAMSTEC, Sasaki et al. 2020) could show lower performances regarding the representation of ocean characteristics variability, but are free to produce their own physics, making those simulations relevant to study specific ocean processes, for example interocean straits exchanges.

Following this comment, we first better explained in the Introduction the importance of simulating realistically small scale processes, both at temporal and spatial scales, including tides, for the study of SCS dynamics. These features were not represented in most of the previous numerical studies of SCS water volume, heat and salt budgets, that used models not including tides and using resolution coarser than 10 km: we highlighted the role of small scale topographic features, especially at interocean straits, of submesoscale to mesoscale dynamics, of tides and induced

mixing, which play a key role in the transformation and transport of water masses through the SCS: lines 72-73 and lines 114-126.

Second, we retrieved four datasets produced from other ocean models, three global simulations (two with assimilation) and one regional simulation (without assimilation):

- CMEMS Global Ocean Physics Analysis and Forecast at 1/12° resolution (~9 km over the SCS region) available over the period 1993-now.

- CMEMS global ocean eddy-resolving reanalysis GLORYS12v1, at 1/12° resolution available over the period 1993-2020.

- OFES (OGCM for the Earth Simulator) version 2 simulation at 1/10° resolution (~11 km) provided by JAMSTEC (Japan Agency for Marine-Earth Science and Technology) over the period 1958-2016, that does not contain assimilation.

- INDESO simulation performed by CLS over the Southeast Asia region at 1/12° over the period 2009-2016, that does not contain assimilation.

We included a description of those coarser resolution simulations in section *2.4 Other global and regional models* and Table 1 of the revised paper. We then included those simulations when comparing our model results with observations data in section 4: we show over 2010-2016 (the period common to all simulations) the time series of climatological monthly mean and interannual yearly mean of SST, SSS and SLA (Figure 5 of the revised paper), the maps of SST, SSS and SLA bias compared to data for the winter and summer period (Figure 7 of the revised paper), the T and S profiles and seasonal cycle of MLD (Figure 10 of the revised paper) and provide the associated values of bias, RMSE and correlations in Table 3 of the revised paper. The performance of SYMPHONIE is compared to the other models in the revised version of the paper in *Section* 4. *Model performance in representing sea surface and water masses characteristics* : lines 506-515 in section *4.1 Sea Surface Characteristics / 4.1.4 Comparison with other models,* lines 537-545 in *4.2 Water masses characteristics,* lines 583-588 in *4.3 Mixed layer depth*. Those comparisons show that the performance of our high resolution simulation in terms of spatial and temporal variability of sea surface characteristics, water masses characteristics and mixed layer depth is in the upper range of the 5 simulations. In particular, our model performs as well, and sometimes better, as models that include assimilation. We also mentioned this comparison in the short summary (lines 15-16), abstract (lines 25-27), introduction (lines 142-145) and conclusion (lines 788-790).

Last, our model computes online each term (lateral oceanic fluxes, surface atmospheric fluxes, river discharges and internal variations) of the water volume, heat and salt budgets. Using available (re)analysis to study those budgets indeed requires to compute them offline, based on daily, weekly or even monthly distributed outputs, thus neglecting the turbulent term of temperature and salinity lateral transports. This is now clearly stated in the introduction (lines 112-114), and explained and assessed in detail in Sectio*n 3 Added-value of the online budget computation.* With offline computation based on daily outputs, NRMSE reaches 10 to 30% for interannual variations of

yearly values of heat and salt net lateral fluxes. Moreover, the online method allows to rigorously compute at each lateral strait the total inflowing and outflowing fluxes, contrary to the offline method that induces errors of the same order or even one order of magnitude larger than the values themselves (see Figure 3 and Table 2 of the revised paper).

2) The circulations in the SCS have not been adequately validated. For instance, the authors mention the significance of the SCS Throughflow, which plays a predominant role in defining the SCS circulation. It is crucial to further validate whether the intensity and structure of this flow are accurately represented in the simulation.

Following this comment, and the comment of the other reviewer, we added a whole section about the evaluation and analysis of water volume budget over the domain, examining the contribution of lateral fluxes at the six interocean straits (Taiwan, Luzon, Mindoro, Balabac, Karimata and Malacca), i.e. the SCSTF, of rivers and of atmosphere: section *5 Evaluation and analysis of SCS interocean straits water volume exchanges and SCS water budget,* pages 32 to 42.

We first presented a synthesis of the observational and numerical estimates available from previous studies (section 5.1 and Table 4 of the revised paper). We then examined the climatological average and seasonal cycle (section 5.1.1, Figure 11) as well as the vertical structure (section 5.1.2, Figure 12) of interocean lateral fluxes of water. To summarize our results explained in detail in section 5.2, we showed that our model reproduces realistically the interocean water volume exchanges in terms of climatological average, seasonal variability and vertical structure.

Finally we examined the contributions of atmosphere and rivers in the water volume budget in section 5.2 and Figures 11. To summarize our results explained in detail in section 5.2, the SCS receives on average a 4.5 Sv yearly water volume input, mainly from the Luzon Strait. It laterally releases this water to neighboring seas, mainly to the Sulu Sea through the Mindoro Strait (49%), to the East China Sea via the Taiwan Strait (28%) and to the Java Sea through the Karimata Strait (22%). The seasonal variability of this water volume budget is driven by lateral interocean exchanges, that largely exceed atmospheric gains or losses and river gains.

We modified the short summary (lines 17-20), abstract (lines 31-38), introduction (lines 145-147) and conclusion (lines 802-830) accordingly.

For the sake of conciseness, and since the paper is already long enough, similar analysis for budgets of heat and salt and analysis of interannual variability will be presented in a future paper, as explained in the conclusion (lines 831-840).

3) The simulation covers the period from 2009 to 2018, and the discussion also focuses on this period. Why was there no mention of the simulation requiring time to spin-up to eliminate distortions caused by abruptly imposed forcings?

We thank the reviewer for pointing out this issue. The spin-up of a model involves two time scales: the physical spin-up time scale and the numerical spin-up time scale.

The physical spin-up scale is very long, of the order of several to tens of years depending on the size of the domain. The goal is to establish ocean circulation from an initial state at rest (i.e. zero current). For example, simulations done with the NEMO model at 1/12° over the Mediterranean Sea, i.e. a domain of comparable size, depart at rest from the climatology and apply a 10 year spin-up to activate the Mediterranean circulation (see Waldman et al. 2018). However, this spin-up does not apply in our case since we don't depart from rest: it only concerns the CMEMS analysis (that uses NEMO at 1/12° over the global ocean), which provides the initial state (including currents) and lateral boundary conditions for our model.

The numerical spin-up scale mainly concerns the adjustment of the initial physical fields to the specific constraints of our grid, for example its bathymetry, which is not exactly the same in our model and in CMEMS analysis, although it is close since it is constructed from the same GEBCO database. Moreover, because of the difference in horizontal resolution, the spectrum of wavelengths represented by our grid is slightly broader than that of the CMEMS analysis. There is therefore a physical spin-up at short wavelengths (those represented by our grid but not by the CMEMS grid). This spin-up lasts for a few months, as can be seen on Figure A below that shows the integral of kinetic energy over the computational domain. In the revised version of the paper, we therefore removed year 2009 from the simulation, and analysed it between 01 January 2010 and 31 December 2018. Note that the results and conclusions were not significantly impacted by the removal of the first year of computation: see for example Table A below where we show the comparison between SYMPHONIE outputs and SST, SSS and SLA observations in average over the domain.

Following this comment, we added a paragraph in section 2.1.2 to explain this (lines 180-183), performed all our analysis (evaluation and computation of fluxes) over the period 2010-2018 (or 2010-2016 when comparing with data or other models that were not available after 2016), and modified the figures and text accordingly throughout the revised paper.

[Figure]

Figure A: Kinetic energy (sum of the square root of u and v - velocity following x and y axis) over the computational domain for the first two year of simulation (2009 - 2011)

Table A : Mean bias, correlation coefficient and NRMSE (for the monthly climatological cycle and interannual time series of yearly average) in SYMPHONIE compared to data (OSTIA for SST, SMOS for SSS and altimetry for SLA) over the periods 2009-2018 and 2010-2018.

| Models | Bias | | | Correlation coefficient R annual cycle / *interannual* | | | NRMSE (%) annual cycle / *interannual* | | |
|---|---|---|---|---|---|---|---|---|---|
| | SST (°C) | SSS | SLA (m) | SST | SSS | SLA | SST | SSS | SLA |
| SYMPHONIE 2009-2018 | -0.18 | -0.04 | 8.6E-5 | 0.99 p<0.01 *0.94 p<0.01* | 0.91 p<0.01 *0.91 p<0.01* | 0.97 p<0.01 *0.88 p<0.01* | 5.71 *27.3* | 18.9 *20.0* | 12.0 *18.5* |
| SYMPHONIE 2010-2018 | -0.18 | -0.04 | -4.5E-4 | 0.99 p<0.01 *0.94 p<0.01* | 0.91 p<0.01 *0.91 p<0.01* | 0.97 p<0.01 *0.90 p<0.01* | 5.73 26.2 | 18.9 20.0 | 9.92 17.5 |

4) The computational domain does not include the source region of the Kuroshio current (e.g. the NEC), which extensively intrudes into the SCS through Luzon Strait. Consequently, the dynamics of this important western-boundary current are not adequately addressed. It would be valuable to assess whether the intensity of the Kuroshio intrusion in the Luzon Strait aligns with observations of volume transport.

Following this comment and the previous comment 2), we added a whole section (Section 5, pages 32-42) about the evaluation of water budget of the SCS and of its components, including the six interocean straits, in particular the Luzon Strait. The climatological values, seasonal cycle and vertical structure of fluxes at the interocean straits is addressed in section 5.1 (*Evaluation of water fluxes at interocean straits,* pages 33-41*)* that includes 5.1.1 (*Climatological mean values and seasonal cycle,* pages 37-38*)* and 5.1.2 (*Vertical structure,* pages 38-41*).* To summarize our results explained in detail in section 5.1, we showed in particular that our model reproduces realistically the interocean water volume exchanges at Luzon Strait in terms of climatological average, seasonal variability (lines 630-643) and vertical structure (lines 694-706). Surface interocean exchanges at Luzon Strait are driven by monsoon winds which favor winter southwestward flows and summer northeastward surface flows. Exchanges through Luzon Strait deep layers show a stable sandwiched structure with vertically alternating inflows and outflows.

We modified the abstract (lines 31-32 and 35-38) and conclusion (lines 807-808 and 827-828) accordingly.

5) Among the widely used numerical simulation kernels, mass conservation is typically replaced by volume conservation under the incompressible assumption. Therefore, volume should be conserved in the computational domain, while salinity and temperature may not be conserved due to additional sources and sinks. I would appreciate it if the authors could explain why "The variation of heat content HC between times t1 and t2 (ΔHC) is equal to the sum of all heat fluxes exchanged within the SCS domain between t1 and t2" is used for heat balance.

Additionally, why are evaporation-precipitation (E-P) and river discharge excluded from the computation of salt flux? Changes in salinity resulting from E-P and river discharge significantly affect the salinity and, consequently, the budget presented in the manuscript.

The SYMPHONIE model is based on the Navier–Stokes primitive equations under the hydrostatic equilibrium hypothesis, incompressibility hypothesis and Boussinesq approximations (see Marsaleix et al. 2008). Similarly as in other ocean models (e.g. NEMO, ROMS, HYCOM), the discretization of equations ensures the conservation of volume, heat and salt contents. This does not mean that the total volume, temperature and salinity integrated over the domain does not change, but means that during every time step, the variation of volume, heat and salt content over the numerical ocean domain is rigorously equal to the net sum of volume, heat and salt input (sources) and output (sinks) at the boundaries of the domain. This is indeed what we obtain when we plot the net sum of inputs and inputs and the variations : see Figure 2 a,b,c of the revised paper. For water volume and heat, lateral ocean boundaries and atmosphere can be sources and/or sinks, and rivers are sources: precipitation and evaporation, shortwave, longwave, latent and sensible heat fluxes at the air-sea interface, river discharge. For salt, the only source/sink is lateral boundary conditions. The salinity of water going to or coming from to the atmosphere and the rivers is indeed assumed to be zero, i.e. there is no input or output of salt from surface atmospheric fluxes and river runoff since rainwater, river water and evaporated water do not contain salt. It should be noted, however, that evaporation, precipitation and river discharge are not sources/sinks of salt, but are sources/sinks of salinity for the ocean domain: although they do not affect the salt budget of the ocean domain, atmospheric and river fluxes do modify the salinity budget, as they affect the water volume budget. Assuming for example a closed domain with no open oceanic lateral boundaries, precipitation and river discharge alone would result in an increase of the total volume with no variation of the total salt content, i.e. in a decrease of salinity. Similarly evaporation alone would result in a decrease of water volume with no variation of salt content, i.e. an increase of salinity.

Following this comment, we added text in section 2.2.1 to better explain this (lines 206-209 and lines 227-231).

6) The simulation was forced with Harmonic Constants from FES2014b, and then the simulated heat content (HC) was compared to FES2014b itself. I may suggest the authors collect record of tidal elevation from tidal gauges and conduct validation.

FES2014b tidal solution is produced from the FES (Finite Element Solution) tidal model. FES2014b assimilates altimetry and tide gauges data (see Carrère et al., 2016 and https://www.aviso.altimetry.fr/en/data/products/auxiliary-products/global-tide-fes/description-fes2014.html), which allows it to reach an unprecedented level of precision and to show accuracy that is superior to the previous versions, in particular to versions without assimilation. We therefore use FES2014b to provide harmonic constants to our simulation, but only at the lateral boundaries of the numerical domain, which are located outside the SCS. We can moreover use it as a reference to evaluate the tidal solution produced by the model over the inner domain, as shown by Piton et al. (2020) over the Gulf of Tonkin. In the open sea, FES2014b is indeed very close to altimetry.

Complementary to that, we agree that comparing our results with tide gauges data is very relevant for the coastal area. We therefore retrieved tide gauges data available from GESLA3.0 (Haigh et al. 2023). Comparing our results with those data and with FES2014b confirms that SYMPHONIE reproduces realistically the tidal solution both over the coast (see Figure 3 of the revised paper) as well as in the open sea (see Figure 4 of the revised paper).

Following this comment, we described tide gauges data in Section *2.3.2 In-situ data*, included the comparison with tide gauges data in section *4.1.1 Tides* (lines 378-385 and Figure 3) and added some text to explain the way we use FES2014b complementary to tide gauges data (lines 390-393).

**References**

Carrère, L., Lyard, F., Cancet, M., Guillot, A., Picot, N.: FES2014, a new tidalmodel - Validation results and perspectives for improvements, presentationto ESA Living Planet Conference, Prague, 2016

Haigh, I. D., Marcos, M., Talke, S. A., Woodworth, P. L., Hunter, J. R., Hague, B. S., Arns, A., Bradshaw, E., and Thompson, P.: GESLA Version 3: A major update to the global higher-frequency sea-level dataset, Geosci Data J, 10, https://doi.org/10.1002/gdj3.174, 2023.

Marsaleix, P., Auclair, F., Floor, J. W., Herrmann, M. J., Estournel, C., Pairaud, I., and Ulses, C.: Energy conservation issues in sigma-coordinate free-surface ocean models, Ocean Model (Oxf), 20, 61–89, https://doi.org/10.1016/j.ocemod.2007.07.005, 2008.

Piton, V., Herrmann, M., Lyard, F., Marsaleix, P., Duhaut, T., Allain, D., and Ouillon, S.: Sensitivity study on the main tidal constituents of the Gulf of Tonkin by using the frequency-domain tidal solver in T-UGOm, Geosci. Model Dev., 13, 1583–1607, https://doi.org/10.5194/gmd-13-1583-2020, 2020.

Sasaki, H., Kida, S., Furue, R., Aiki, H., Komori, N., Masumoto, Y., Miyama, T., Nonaka, M., Sasai, Y., and Taguchi, B.: A global eddying hindcast ocean simulation with OFES2, Geosci. Model Dev., 13, 3319–3336, https://doi.org/10.5194/gmd-13-3319-2020, 2020.

Yang, Y., Wang, D., Wang, Q., Zeng, L., Xing, T., He, Y., et al.: Eddy-induced transport of saline Kuroshio water into the northern South China Sea. Journal of Geophysical Research: Oceans, 124, 6673– 6687. https://doi.org/10.1029/2018JC014847, 2019.

Zhitao Yu, E. Joseph Metzger, Yalin Fan.: Generation mechanism of the counter-wind South China Sea

Warm Current in winter, Ocean Modelling, Volume 167, 101875, ISSN 1463-5003, https://doi.org/10.1016/j.ocemod.2021.101875, 2021.

Waldman R., S. Somot, M. Herrmann, F. Sevault, P.E. Isachsen.: On the chaotic variability of deep convection in the Mediterranean Sea. Geophys. Res. Let., https://doi.org/10.1002/2017GL076319, 2018

---

## Author Comment (AC2)

**Answer to reviewer 2**

This manuscript presents a study on the implementation and evaluation of a high-resolution hydrodynamic model (SYMPHONIE) over the South China Sea (SCS). The authors simulate a 10-year period and successfully replicate observed circulation patterns and water masses in the SCS. The introduction of an online computation method to assess water volume, heat, and salt budgets adds strength to the manuscript. However, there are some areas that require improvement. First, the manuscript should clearly highlight the role and advantages of the 4 km configuration, enhancing the scientific significance of the study. Second, Section 3 needs restructuring to ensure logical flow and provide a more detailed evaluation of the South China Sea Throughflow (SCSTF). Lastly, reconsidering the title to clarify the "scale" aspect would be beneficial. By addressing these issues, the manuscript has the potential to make a valuable contribution to the scientific literature on South China Sea dynamics and ocean modeling community.

We warmly thank the reviewer for the time and attention devoted to our paper, and for those positive and constructive comments. We have carefully considered all the comments and suggestions in the revised version of our manuscript. In what follows, and in the highlighted version of the manuscript, our answers and modifications are highlighted in blue. Line numbers refer to the highlighted version of the revised manuscript.

Major Comments:

1. The title of this manuscript is confusing and could be misunderstood as referring to different spatial scales of ocean dynamics rather than temporal scales. It is recommended to clarify this in the title to avoid confusion.

The initial title was "Studying multi-scale ocean dynamics and their contribution to water, heat and salt budgets in the South China Sea : evaluation of a high-resolution configuration of an online closed-budget hydrodynamical ocean model (SYMPHONIE version 249). Following this comment, as well as the modification done in the revised paper that now includes an evaluation and analysis of water volume budget over the SCS, we changed the title to "**New insights on the South China Sea Throughflow and its seasonal cycle: evaluation and analysis of a configuration at high-resolution including tides of the online closed-budget regional ocean model SYMPHONIE version 2.4**"

2. The manuscript mentions the use of a 4 km resolution, which is an outstanding feature of the configuration. It would be more meaningful to explicitly highlight the role and advantages of this high-resolution approach in the manuscript. This will enhance the significance of the "multi-scale ocean dynamics" concept, incorporating both temporal and spatial scales.

First, following this comment, and based on the literature, we better explained in the Introduction the importance of simulating realistically small scale processes, both at temporal and spatial scales, including tides, for the study of SCS dynamics. These features were not represented in most of the

previous numerical studies of SCS water volume, heat and salt budgets, that used models not including tides and using resolution coarser than 10 km: we highlighted the role of small scale topographic features, especially at interocean straits, of submesoscale to mesoscale dynamics, of tides and induced mixing, which play a key role in the transformation and transport of water masses through the SCS: lines 72-73 and lines 114-126.

We moreover quantitatively showed the added-value of our high-resolution simulation compared to simulations at coarser resolution: see the answer to the following comment.

Last, our model computes online each term (lateral oceanic fluxes, surface atmospheric fluxes, river discharges and internal variations) of the water volume, heat and salt budgets. Using available (re)analysis to study those budgets indeed requires to compute them offline, based on daily, weekly or even monthly distributed outputs, thus neglecting the turbulent term of temperature and salinity lateral transports. This is not exactly an advantage of the high-resolution, but it is an advantage of our configuration compared to other products. This is now clearly stated in the introduction (lines 112-114) and explained and assessed in detail in Sectio*n 3 Added-value of the online budget computation.* With offline computation based on daily outputs, NRMSE reaches 10 to 30% for interannual variations of yearly values of heat and salt net lateral fluxes. Moreover, the online method allows to rigorously compute at each lateral strait the total inflowing and outflowing fluxes, contrary to the offline method that induces errors of the same order or even one order of magnitude larger than the values themselves (see Figure 3 and Table 2 of the revised paper).

3. The authors claim that the 4 km configuration in the South China Sea (SCS) is developed, but it is important to explain the advantages of this resolution compared to coarser resolution models. Describing these advantages will enhance the scientific significance of the manuscript.

Several groups indeed develop and distribute global or regional simulations that cover the SCS region from other models. Some of those simulations (for example reanalysis and analysis produced by CMEMS and most of HYCOM simulations used to study the area, e.g. Yang et al. 2019, Zhitao et al. 2021) include assimilation procedures toward satellite sea surface temperature and elevation data and ARGO temperature and salinity profiles. This helps them to realistically reproduce ocean surface characteristics and water masses profiles as well as their variability, but does not let them completely free to produce their own physics. Conversely, simulations without assimilation (e.g OFES simulations produced by JAMSTEC, Sasaki et al. 2020) could show lower performances regarding the representation of ocean characteristics variability, but are free to produce their own physics, making those simulations relevant to study specific ocean processes, for example interocean straits exchanges.

Following this comment, we retrieved four datasets produced from other ocean models, three global simulations (two with assimilation) and one regional simulation (without assimilation):

- CMEMS Global Ocean Physics Analysis and Forecast at 1/12° resolution (~9 km over the SCS region) available over the period 1993-now.

- CMEMS global ocean eddy-resolving reanalysis GLORYS12v1, at 1/12° resolution available over the period 1993-2020.

- OFES (OGCM for the Earth Simulator) version 2 simulation at 1/10° resolution (~11 km) provided by JAMSTEC (Japan Agency for Marine-Earth Science and Technology) over the period 1958-2016, that does not contain assimilation.

- INDESO simulation performed by CLS over the Southeast Asia region at 1/12° over the period 2009-2016, that does not contain assimilation.

We included a description of those coarser resolution simulations in section *2.4 Other global and regional models* and Table 1 of the revised paper.  We then included those simulations when comparing our model results with observations data in section 4 : we show over 2010-2016 (the period common to all simulations) the time series of climatological monthly mean and interannual yearly mean of SST, SSS and SLA (Figure 5 of the revised paper), the maps of SST, SSS and SLA bias compared to data for the winter and summer period (Figure 7 of the revised paper), the T and S profiles and seasonal cycle of MLD (Figure 10 of the revised paper) and provide the associated values of bias, RMSE and correlations in Table 3 of the revised paper. The performance of SYMPHONIE is compared to the other models in the revised version of the paper in *Section* 4. *Model performance in representing sea surface and water masses characteristics* : lines 506-515 in section *4.1 Sea Surface Characteristics / 4.1.4 Comparison with other models,* lines 537-545 in *4.2 Water masses characteristics,* lines 583-588 in *4.3 Mixed layer depth*. Those comparisons show that the performance of our high resolution simulation in terms of spatial and temporal variability of sea surface characteristics, water masses characteristics and mixed layer depth is in the upper range of the 5 simulations. In particular, our model performs as well, and sometimes better, as models that include assimilation. We also mentioned this comparison in the short summary (lines 15-16), abstract (lines 25-27), introduction (lines 142-145) and conclusion (lines 788-790).

4. The structure of Section 3 is unclear. The title of Section 3.2 ("surface characteristics") overlaps with the title of Section 3.1 ("tide"), and the title of Section 3.2.3 is inconsistent with other subsection titles. It is recommended to restructure Section 3 to make it more coherent and clear.

Following this comment, and taking into account the other comments of both reviewers, we modified the structure of the paper to improve its clarity: after presenting the methods and data in section 2, we present the added-value of the online computation of budgets in section 3. We then compared our results with available satellite and in-situ data as well as other coarser resolution simulation in section 4. As explained below (comment 6), a whole section 5 was then added to evaluate the representation of water budget and SCSTF.

To make the structure of section 4 clearer, it is now divided in *4.1 Sea surface characteristics*, with *4.1.1 Tides*, *4.1.2 Seasonal cycle of sea surface temperature, salinity and elevation 4.1.3,*

*Interannual variations of sea surface temperature, salinity and elevation,* then *4.2 Water masses characteristics* and *4.3 Mixed Layer Depth.*

5. The analysis of Mixed Layer Depth (MLD) in Section 3.4 is interesting, revealing a robust shallower bias due to wind speed. However, considering the higher resolution, a deeper MLD might be expected, particularly in winter. It would be valuable to compare the results of the 4 km configuration with a coarser-resolution configuration.

Following this comment, we included the 4 other simulations mentioned above (comment 3, OFES, INDEOSO, COPERNICUS and GLORYS) when comparing the simulated MLD with MLD computed from ARGO in Section *4.3 Mixed Layer De*pth: see Figure 10, Table 3 and lines 583-588 of the revised paper.

All models indeed underestimate MLD (certainly due to the same reason, i.e. underestimation of wind speed in atmospheric forcing datasets), but simulate similar annual evolution of MLD. SYMPHONIE performance is in the upper range: OFES shows the smallest underestimation and NRMSE (7.3 m, 19%), followed by SYMPHONIE (9.1 m, 25%), then INDESO (10.5 m, 28%). The strongest biases and NRMSE are obtained from INDESO (15.6 m, 41%) and COPERNICUS (15.5 m, 40%), even if those models assimilate Argo profiles.

6. Since the South China Sea Throughflow (SCSTF) is mentioned as a major topic at the beginning of the manuscript, a detailed evaluation of all inflows and outflows through different straits would be expected. However, Section 3 (model evaluation) does not address these throughflows. It is recommended to include an evaluation of the SCSTF in Section 3.

Following this comment, and the comment of the other reviewer, we added a whole section about the evaluation and analysis of water volume budget over the domain, examining the contribution of lateral fluxes at the six interocean straits (Taiwan, Luzon, Mindoro, Balabac, Karimata and Malacca), i.e. the SCSTF, of rivers and of atmosphere: section *5 Evaluation and analysis of SCS interocean straits water volume exchanges and SCS water budget,* pages 32 to 42.

We first presented a synthesis of the observational and numerical estimates available from previous studies (section 5.1 and Table 4 of the revised paper). We then examined the climatological average and seasonal cycle (section 5.1.1, Figure 11) as well as the vertical structure (section 5.1.2, Figure 12) of interocean lateral fluxes of water based on the SYMPHONIE simulation. To summarize our results explained in detail in section 5.1, we showed that our model reproduces realistically the interocean water volume exchanges in terms of climatological average, seasonal variability and vertical structure. Surface interocean exchanges, especially at Luzon Strait, are all driven by monsoon winds which favor winter southwestward flows and summer northeastward surface flows. Exchanges through Luzon Strait deep layers show a stable sandwiched structure with vertically alternating inflows and outflows.

Finally, we examined the contributions of atmosphere and rivers in section 5.2 and Figure 11. To summarize our results explained in detail in section 5.2, the SCS receives on average a 4.5 Sv

yearly water volume input, mainly from the Luzon Strait. It laterally releases this water to neighbouring seas, mainly to the Sulu Sea through the Mindoro Strait (49%), to the East China Sea via the Taiwan Strait (28%) and to the Java Sea through the Karimata Strait (22%). The seasonal variability of this water volume budget is driven by lateral interocean exchanges, that largely exceed atmospheric gains or losses and river gains.

We modified the short summary (lines 17-20), abstract (lines 31-38), introduction (lines 145-147) and conclusion (lines 802-830) accordingly.

For the sake of conciseness, and since the paper is already long enough, similar analysis for budgets of heat and salt and analysis of interannual variability will be presented in a future paper, as explained in the conclusion (lines 831-840).

Minor Comments:

1. Maintain consistency in the expression of water volume, using consistent terminology throughout the manuscript (e.g., water, volume, or water volume).

→ We chose "water volume" and carefully checked that we used this terminology consistently throughout the manuscript.

2. In Line 218, consider including river fluxes in Section 2.2.2 (lateral fluxes) for better organization.

→ The term "lateral fluxes" actually refers to fluxes through lateral open ocean boundaries, i.e. interocean straits. We corrected our text to make things clearer: in the revised manuscript, we kept separate the sections for lateral oceanic fluxes and river fluxes, since they are computed differently, but added the word "oceanic" with "lateral" throughout the text and moved the section about river fluxes (now 2.2.3) just after the section about lateral oceanic fluxes (now 2.2.2) for consistency.

3. In Figure 2, it is difficult to discern the difference between the simulation and tidal product. Including a column for bias would enhance clarity.

→ Done : we added bias maps in Figures 3 (tide gauges) and 4 (FES2014b).

4. Figure 4 should include the bias information. Adding the bias to the figure will improve interpretation.

→ Done : we added bias maps in Figure 7.

5. In Lines 312, 314, 361, 408, 479, etc., add units for the variables.

→ Units (psu) are added everywhere for salinity in the revised manuscript.

6. In Line 396, consider revising to "Therefore, our simulation accurately represents..."

→ Done (lines 534-535)

7. The numbering in the caption of Figure 7 is incorrect, as "(c)" is used twice. Adjust the numbering accordingly.

→ Indeed, this was corrected. We have verified and adjusted the numbering of all figures in the revised manuscript

8. Maintain consistency in formatting, ensuring that there is either a space or no space between paragraphs throughout the manuscript.

→ We have adjusted the formatting, ensuring that there is a space between each paragraph.

**References**

Sasaki, H., Kida, S., Furue, R., Aiki, H., Komori, N., Masumoto, Y., Miyama, T., Nonaka, M., Sasai, Y., and Taguchi, B.: A global eddying hindcast ocean simulation with OFES2, Geosci. Model Dev., 13, 3319–3336, https://doi.org/10.5194/gmd-13-3319-2020, 2020.

Yang, Y., Wang, D., Wang, Q., Zeng, L., Xing, T., He, Y., et al.: Eddy-induced transport of saline Kuroshio water into the northern South China Sea. Journal of Geophysical Research: Oceans, 124, 6673– 6687. https://doi.org/10.1029/2018JC014847, 2019.

Zhitao Yu, E. Joseph Metzger, Yalin Fan.: Generation mechanism of the counter-wind South China Sea Warm Current in winter, Ocean Modelling, Volume 167, 101875, ISSN 1463-5003, https://doi.org/10.1016/j.ocemod.2021.101875, 2021.

---

## Referee Report (RR1)

**Reviewer #2**

I recommend that the manuscript be accepted for final publication after the authors address the remaining concerns and suggestions from the reviewers. The authors have made significant improvements in response to the initial major comments and have effectively addressed issues related to the title, advantages of their high-resolution approach, restructuring of sections, and comparisons with other models. The addition of a dedicated section on the South China Sea Throughflow (SCSTF) and water budget analysis further enhances the manuscript's content.

However, there are a few questions and recommendations that should be addressed in the final revision. The authors should provide more clarity regarding the underestimation of Mixed Layer Depth (MLD) and the role of wind speed, as well as the robustness of lateral interocean exchanges ratios and their validation against observational or previous studies. Additionally, the suggestion to use a coarser version of the same model core for comparison instead of other coarser-resolution models should be addressed if possible, exhibiting the benefits of the higher-resolution more directly. Once these remaining points are adequately addressed, the manuscript should be considered suitable for final publication.

**Major Comments:**
1. MLD Underestimation and Wind Speed: How the authors concluded that all models' underestimation of Mixed Layer Depth (MLD) is attributed to the underestimation of wind speed. Do all models use the same forcing? It's important for the authors to clarify and provide further details on the role of wind speed and the consistency of forcing among the models when discussing MLD underestimation.

2. Use of Coarser Configuration: It is recommended that the authors consider using a coarser version of the same model core for comparison, in addition to comparing their high-resolution model with other coarser-resolution models. Including a coarser configuration with the same model core will provide a more convincing basis for highlighting the benefits of their new setting, especially in terms of finer resolution and other relevant factors. This approach can offer a more direct and robust assessment of the advantages of their high-resolution model.

3. Robustness of Lateral Interocean Exchanges Ratio: The ratio of different lateral interocean exchanges is interesting. I am also wondering if it is robust. If there is any observational or previous study support for this ratio. It's essential for the authors to provide information on the robustness and potential sources of validation for the ratios presented in the study.

**Minor Comments:**
1. Title Prolixity: The new title appears somewhat lengthy. It might be advisable to streamline the title by removing some of the detailed configuration information, thus achieving a more concise and reader-friendly title. The authors should contemplate this suggestion and determine if certain elements of the title can be omitted while retaining the core information.

2. Legend of Figure 2: The legend of Figure 2c is overlapped with the graph.

---

## Referee Report (RR2)

The revised version of the manuscript has undergone significant improvements in response to previous rounds of review. The authors have diligently addressed major concerns raised by the reviewers, resulting in a substantially enhanced scientific contribution. The underestimation of Mixed Layer Depth (MLD) and its correlation with wind speed, introducing clarifications on the use of bulk formulae and incorporating additional information on wind speed underestimation. The inclusion of the SYM12 simulation and detailed comparisons with SYM4 have strengthened the study's assessment of the benefits of high-resolution modeling. The robustness of the lateral interocean exchanges ratio has been reasonably explained, and numerical estimates were added to address concerns about the lack of in-situ observations. Minor revisions, such as a concise title and corrected figure legend, further enhance the manuscript's overall quality.

In conclusion, I recommend the acceptance of this manuscript for final publication. The authors have successfully addressed all major and minor concerns, resulting in a well-polished, scientifically sound, and valuable contribution to our understanding of the South China Sea Throughflow and water budget seasonal cycle. The study's methodology, analyses, and conclusions are robust, and the manuscript is ready for dissemination to the scientific community.

---

## Author Response (AR2)

To :

Dr. Qiang Wang, Editor of GMD

From:

Trinh Bich Ngoc

USTH, Hanoi, Vietnam

Marine Herrmann, Caroline Ulses, Patrick Marsaleix, Thomas Duhaut, Claude Estournel

LEGOS, Toulouse, France

To Duy Thai

IO, Nha Trang, Vietnam

R. Kipp Shearman

Oregon State University, Corvallis, United States

December 9, 2023

Dear Editor,

Please find a revised version of our paper entitled "***New insights on the South China Sea Throughflow and water budget seasonal cycle : evaluation and analysis of a high-resolution configuration of the ocean model SYMPHONIE version 2.4***".

We warmly thank the reviewers for their careful reading and their constructive evaluation of our manuscript. We sincerely believe that these remarks helped us to improve and clarify the manuscript. We carefully took their comment into account.

In particular:

- We performed a twin simulation at coarser (12 km) horizontal resolution to further highlight the advantages of the high-resolution and investigate the influence of this resolution on the water budget analysis.

- We investigated more into details the factors for the mixed layer depth underestimation.

- We compared our analysis of the ratios of interocean straits exchanges with previous studies.

Best regards,

Trinh Bich Ngoc and Marine Herrmann

**Answer to reviewers' comments**

We warmly thank both reviewers for the time and attention devoted to our paper, and for those positive and constructive comments. We have carefully considered all the comments and suggestions in the revised version of our manuscript. In what follows, and in the highlighted version of the manuscript, our answers and modifications are highlighted in blue. Line numbers refer to the highlighted version of the revised manuscript.

**Answers to reviewer 1:**

The authors have taken great care to address my concerns in the latest round of review, incorporating extensive validations of the simulation. It is compelling that this simulation effectively elucidates the general circulations in the South China Sea. Therefore, I highly recommend its acceptance.

The manuscript's writing and logic remain sound. I have only two minor suggestions.

1. Firstly, in Figure 6, the representation of the SCS basin appears slightly distorted.

   → this was corrected : we adjusted the Figure 6

2. Secondly, in Table 4, it would be beneficial to reorganize the citations, perhaps by including a line before Wyrtki's pioneer work to indicate that the subsequent citations primarily pertain to observational studies.

   → Following this comment, we added a line in the Table 4 before Wyrtki's pioneer work to indicate the citations of observational studies

3. Furthermore, the examination of volume transport in the Taiwan Strait could be further fortified by incorporating references such as Hu et al. (2010) and some more recent reviews. This would be particularly sensible given the better sampling of this shallow strait compared to others surrounding the SCS basin. Of course, an estimate of about 1.2 Sv remains reasonable.

   → Following this comment, we added the review study of *Hu et al. (2010)* and *Isobe et al. (2008)* in the Table 4. These studies are also cited in the discussion part on Taiwan strait (lines 672-680)

**References**

Hu, J., Kawamura, H., Li, C., Hong, H., and Jiang, Y.: Review on current and seawater volume transport through the Taiwan Strait, https://doi.org/10.1007/s10872-010-0049-1, 2010.

Isobe, A.: Recent advances in ocean-circulation research on the Yellow Sea and East China Sea shelves, https://doi.org/10.1007/s10872-008-0048-7, 2008.

**Answers to reviewer 2:**

I recommend that the manuscript be accepted for final publication after the authors address the remaining concerns and suggestions from the reviewers. The authors have made significant improvements in response to the initial major comments and have effectively addressed issues related to the title, advantages of their high-resolution approach, restructuring of sections, and comparisons with other models. The addition of a dedicated section on the South China Sea Throughflow (SCSTF) and water budget analysis further enhances the manuscript's content. However, there are a few questions and recommendations that should be addressed in the final revision. The authors should provide more clarity regarding the underestimation of Mixed Layer Depth (MLD) and the role of wind speed, as well as the robustness of lateral interocean exchanges ratios and their validation against observational or previous studies. Additionally, the suggestion to use a coarser version of the same model core for comparison instead of other coarser-resolution models should be addressed if possible, exhibiting the benefits of the higher resolution more directly. Once these remaining points are adequately addressed, the manuscript should be considered suitable for final publication.

Major Comments:

1. MLD Underestimation and Wind Speed: How the authors concluded that all models' underestimation of Mixed Layer Depth (MLD) is attributed to the underestimation of wind speed. Do all models use the same forcing? It's important for the authors to clarify and provide further details on the role of wind speed and the consistency of forcing among the models when discussing MLD underestimation.

All simulations considered in our paper use bulk formulae of *Large and Yeager (2004).* Our SYMPHONIE simulations, INDESO and COPERNICUS use atmospheric outputs from the European Centre for Medium Range Weather Forecasts (ECMWF) operational forecasts (1/8° ~14 km horizontal resolution and 3 hours temporal resolution), and GLORYS uses the outputs of ERA-Interim reanalysis (80 km and 3 hours resolution) produced from the same ECMWF model. OFES uses the atmospheric surface dataset JRA55 atmospheric reanalysis (~55 km and 3 hours, resolution, *Kobayashi et al., 2015*). This information is summarized in Table 1 of the revised manuscript.

Compared to QuikSCAT, ECMWF analysis and reanalysis indeed underestimate sea surface wind speed (by ~1 m.s$^{-1}$ on average over the region for ECMWF analysis, see Fig. SM1). *Herrmann et al. (2020, 2022)* indeed showed that global and regional atmospheric models generally underestimate sea surface wind speed over the SEA region, and *Wang et al. (2020)* showed that both ERA-Interim and JRA55 underestimate observed wind dataset over China Seas, with a smaller bias in JRA55 (0.22 m.s$^{-1}$ over the period 1988-2015) than in ECMWF product (0.62 m.s$^{-1}$). This underestimation of wind speed in forcing datasets partly explains why all models underestimate the MLD, and why OFES, which uses JRA55, produces the closest MLD to observations (Fig. 10f of the revised paper).

Moreover, as shown by *Tréguier et al. (2023),* MLD biases as well as their differences among models may also be due to the models formulation, parameterizations and resolution, whose shortcomings vary between models: horizontal and vertical resolutions, inclusion of tides, vertical mixing parameterisation, advection schemes, etc. *Gaube et al. (2019)* for example showed that

mesoscale eddies, whose representation depends on those formulations, modulate the MLD. Indeed, though SYMPHONIE simulations at 4 km (SYM4) and 12 km (SYM12, see answer to comment 2 below) resolutions and COPERNICUS (which provides the initial and lateral oceanic boundary conditions to SYMPHONIE) use the same atmospheric conditions, the MLD underestimation is lower in SYM4 than in SYM12 and COPERNICUS. This suggests first that the MLD underestimation in SYMPHONIE can also be explained by the MLD underestimation of the COPERNICUS initial profiles and those profiles entering the domain, and second that SYMPHONIE, due to different formulations in particular its highest resolution, is able to partially correct the stratification of these initial and entering profiles.

→ following this comment, we added this analysis in our revised paper (Part 4.3, lines 598-617)

2. Use of Coarser Configuration: It is recommended that the authors consider using a coarser version of the same model core for comparison, in addition to comparing their high-resolution model with other coarser-resolution models. Including a coarser configuration with the same model core will provide a more convincing basis for highlighting the benefits of their new setting, especially in terms of finer resolution and other relevant factors. This approach can offer a more direct and robust assessment of the advantages of their high-resolution model.

We agree that running a simulation with the same model and choices of parameterizations, but a coarser resolution, is indeed a rigorous way to highlight and quantify the added-value of the high-resolution. We therefore performed a simulation, called SYM12 in the revised version of the manuscript, using exactly the same configuration, forcing etc as the simulation at 4 km (now called SYM4), but with a 12 km horizontal resolution. In order to avoid any confusion, the simulation at 4 km is now referred to as SYM4 in the whole document.

→ SYM12 simulation is presented in Part 2.4 (lines 314-317) and Table 1 of the revised manuscript.

We then examined the differences between SYM4, in terms of performance and results.

First, SYM4 performs better than SYM12 for all the diagnostics performed here : tidal amplitude (Figure A3 of the Appendices), spatial (Figure 7 and Table 3 of the revised manuscript) and temporal (Figure 5) variability of SST, SSS and SLA, and TS profiles and MLD (Figure 10). Moreover, biases for SYM12 are in the range of the biases obtained for the other simulations at similar resolution, without (INDESO, OFES) or with (GLORYS, COPERNICUS) assimilation. This shows that the SYMPHONIE model at 12 km resolution shows performances similar to those of models with similar resolution, and that using a higher 4 km resolution significantly improves this performance. This can be attributed to the better representation of (sub)mesoscale processes, of tides, but also of topography, in particular of straits. This is particularly obvious when visualizing the bathymetry for the 4 km and 12 km grids in Figure A1, A2 of the Appendices: deep narrow passages are sometimes represented with no more than 1 grid point, in particular at Luzon, Mindoro and Balabac straits.

→ SYM12 is integrated in the comparison with other datasets in the revised manuscript : for sea surface elevation, temperature and salinity (Table 3, Parts 4.1.1 lines 407-414 and 4.1.4 lines 513-

521, Figures 5 and 7), and water masses characteristics and MLD (Parts 4.2 lines 543-555 and 4.3 lines 595-617 and Figure 10).

Second, the resolution of the model also impacts the results on water fluxes at interocean straits (vertical structure (Figure 12), yearly cycle (Figure 11b) integrated estimates (Figure 11a)) as well as the water budget over the area (Figure 11a,c).

→ this influence on water fluxes estimates is integrated in Part 5.3 (lines 805-823) of the revised manuscript, and SYM12 results are included in Figures 11 and 12.

Last, we added a few sentences in the Abstract (lines 23-25 and lines 37) and Conclusion (lines 840-842 and lines 892-893) to mention this study of model resolution.

3. Robustness of Lateral Interocean Exchanges Ratio: The ratio of different lateral interocean exchanges is interesting. I am also wondering if it is robust. If there is any observational or previous study support for this ratio. It's essential for the authors to provide information on the robustness and potential sources of validation for the ratios presented in the study.

The spatial and temporal coverage of the available in-situ observations, reported in Table 3, does not currently allow these ratios to be estimated. An in-situ estimate would indeed require simultaneous measurements to be taken in all the straits over a long period, which is fairly costly and technically complex. The idea is therefore to use observations to assess the quality of the circulation and water masses produced by the models, and then to use the models to 'fill in the gaps' in the observations and compute ratios. Moreover, it would still be extremely relevant to put in place a strategy for measuring these ratios in situ, which would make it possible, among other things, to assess the robustness of these numerical estimates.

→ following this comment, we added, when possible, the values of ratios between water flows through Luzon, Taiwan, Mindoro and Karimata straits estimated from numerical results in Table 4, and discussed this question in the Conclusion, lines 861-872.

Minor Comments:

1. Title Prolixity: The new title appears somewhat lengthy. It might be advisable to streamline the title by removing some of the detailed configuration information, thus achieving a more concise and reader-friendly title. The authors should contemplate this suggestion and determine if certain elements of the title can be omitted while retaining the core information.

→ Following this comment and the suggestion of the editor, we propose the following title : *New insights on the South China Sea Throughflow and water budget seasonal cycle : evaluation and analysis of a high-resolution configuration of the ocean model SYMPHONIE version 2.4*

2. Legend of Figure 2: The legend of Figure 2c is overlapped with the graph.

→ this was corrected, we adjusted the Figure 2

**References**

Gaube, P., J. McGillicuddy, D., and Moulin, A. J.: Mesoscale Eddies Modulate Mixed Layer Depth Globally, Geophys Res Lett, 46, https://doi.org/10.1029/2018GL080006, 2019.

Herrmann, M., Ngo-Duc, T., and Trinh-Tuan, L.: Impact of climate change on sea surface wind in Southeast Asia, from climatological average to extreme events: results from a dynamical downscaling, Clim Dyn, 54, https://doi.org/10.1007/s00382-019-05103-6, 2020.

Herrmann, M., Nguyen-Duy, T., Ngo-Duc, T., and Tangang, F.: Climate change impact on sea surface winds in Southeast Asia, International Journal of Climatology, 42, https://doi.org/10.1002/joc.7433, 2022.

Kobayashi, S., Ota, Y., Harada, Y., Ebita, A., Moriya, M., Onoda, H., Onogi, K., Kamahori, H., Kobayashi, C., Endo, H., Miyaoka, K., and Kiyotoshi, T.: The JRA-55 reanalysis: General specifications and basic characteristics, Journal of the Meteorological Society of Japan, 93, https://doi.org/10.2151/jmsj.2015-001, 2015.

Large, W. G. and Yeager, S. G.: Diurnal to decadal global forcing for oceans and sea-ice models : the data ses and flux climatologies., https://doi.org/10.5065/D6KK98Q6, 2004.

Tréguier, A. M., De Boyer Montégut, C., Bozec, A., Chassignet, E. P., Fox-Kemper, B., McC Hogg, A., Iovino, D., Kiss, A. E., Le Sommer, J., Li, Y., Lin, P., Lique, C., Liu, H., Serazin, G., Sidorenko, D., Wang, Q., Xu, X., and Yeager, S.: The mixed-layer depth in the Ocean Model Intercomparison Project (OMIP): impact of resolving mesoscale eddies, Geosci Model Dev, 16, https://doi.org/10.5194/gmd-16-3849-2023, 2023.

Wang, G., Wang, X., Wang, H., Hou, M., Li, Y., Fan, W., and Liu, Y.: Evaluation on monthly sea surface wind speed of four reanalysis data sets over the China seas after 1988, Acta Oceanologica Sinica, 39, https://doi.org/10.1007/s13131-019-1525-0, 2020.